# Measuring user interactions with websites: A comparison of two industry standard analytics approaches using data of 86 websites

**Bernard J. Jansen**[1☺]*, **Soon-gyo Jung**[1☺], **Joni Salminen**[1,2,3☺]

**1** Qatar Computing Research Institute, Hamid Bin Khalifa University, Doha, Qatar, **2** Turku School of Economics, University of Turku, Turku, Finland, **3** School of Marketing and Communication, University of Vaasa, Vaasa, Finland

☺ These authors contributed equally to this work.
* jjansen@acm.org

**Data Availability Statement:** The data underlying the results presented in the study are available from SimilarWeb (https://www.similarweb.com/).

## Abstract

This research compares four standard analytics metrics from Google Analytics with Similar-Web using one year's average monthly data for 86 websites from 26 countries and 19 industry verticals. The results show statistically significant differences between the two services for total visits, unique visitors, bounce rates, and average session duration. Using Google Analytics as the baseline, SimilarWeb average values were 19.4% lower for total visits, 38.7% lower for unique visitors, 25.2% higher for bounce rate, and 56.2% higher for session duration. The website rankings between SimilarWeb and Google Analytics for all metrics are significantly correlated, especially for total visits and unique visitors. The accuracy/inaccuracy of the metrics from both services is discussed from the vantage of the data collection methods employed. In the absence of a gold standard, combining the two services is a reasonable approach, with Google Analytics for onsite and SimilarWeb for network metrics. Finally, the differences between SimilarWeb and Google Analytics measures are systematic, so with Google Analytics metrics from a known site, one can reasonably generate the Google Analytics metrics for related sites based on the SimilarWeb values. The implications are that SimilarWeb provides conservative analytics in terms of visits and visitors relative to those of Google Analytics, and both tools can be utilized in a complementary fashion in situations where site analytics is not available for competitive intelligence and benchmarking analysis.

## Introduction

Web analytics is the collection, measurement, analysis, and reporting of digital data to enhance insights concerning the behavior of website visitors [1]. Web analytics is a critical component of business intelligence, competitive analysis, website benchmarking, online advertising, online marketing, and digital marketing [2] as business decisions are made based on website

The authors had no special access privileges to the data others would not have.

**Funding:** The author(s) received no specific funding for this work.

**Competing interests:** The authors have declared that no competing interests exist.

traffic measures obtained from website analytics services. Organizations monitor their sites' incoming and outgoing traffic to identify popular pages, determine user interests, and stay abreast of emerging trends [3]. There are various ways to monitor this traffic, and the gathered data is used for re-structuring sites, highlighting security problems, indicating bandwidth issues, assessing organizational key performance indicators (KPIs), and obtaining societal insights [4].

Approaches to collecting website analytics data can be grouped by the focus of data collection efforts, resulting in the emergence of three general methodologies, namely: (a) *user-centric*, (b) *site-centric*, and (c) *network-centric*. The central traits of each are as follows.

- **User-centric**: Web analytics data is gathered via a panel of users, which is tracked by software installed on users' computers, such as a plugin for a web browser [5–8]. For example, when users install an extension to their browser, they approve in the license agreement that the data on the websites they visit will be processed and analyzed. The primary advantage here is that the user-centric approach does not rely on cookies or tags (i.e., snippets of information placed by a server to a user's web browser in order to keep track of the user) but on direct observation. An additional advantage is comparing web analytics data across multiple websites. The challenge is recruiting and incentivizing a sufficiently large user panel that is a representative sample of the online population—due to this challenge, only a few companies have recruited sizeable user panels (e.g., Alexa). Another disadvantage may be the issue of privacy since many users are not willing to share information on every website that they visit, so some users may make efforts to mask their actual online actions from the tracking plugin.

- **Site-centric**: Web analytics is gathered via software on a specific website [9–16]. Most websites use a site-centric approach for analytics data gathering, typically employing cookies and/or tagging pages on the website (e.g., Google Analytics, Adobe Analytics). The primary advantage of this approach lies in counting events and actions (e.g., pages viewed, times accessed), which is relatively straightforward. Another advantage is that users do not need to install specific software beyond the browser. However, there are disadvantages. First, site-centric software focuses on cookies/tags, so these counts may not reflect actual people (i.e., the measures are of the cookies and tags) or people's actual actions on the website. Instead, site-centric approaches measure the number of cookies dropped or tags fired as proxies for people or interactions. Second, this approach is susceptible to bots (i.e., autonomous programs that pretend to be real users) and other forms of analytics inflation tactics, such as click fraud [17]. Finally, the site-centric analytics usually represent just *one* website and are only accessible to the owner of that website, making the site-centric approach not widely available for business intelligence, marketing, advertising, or other tasks requiring web analytics data from a large number of sites.

- **Network-centric**: Web analytics is gathered via observing and collecting traffic in the network [18, 19]. There are various techniques for network-centric web analytics data gathering, with the most common being data purchased or acquired directly from Internet service providers (ISPs). However, other data gathering methods include leveraging search traffic, search engine rankings, paid search, and backlinks [20, 21]. The main advantage of the network-centric approach is that one can relatively easily collect analytics concerning a large number of websites. Also, the setup is comparatively easy, as neither users nor websites are required to install any software. The major disadvantage is that there is no information about the onsite actions of the users. A second disadvantage is that major ISPs do not freely share their data, so acquiring it can be expensive. However, companies can acquire other

network-centric data more reasonably (i.e., SpyFu, SEMRush; two common industry tools for search marketing), albeit requiring substantial computational, programming, and storage resources.

Of course, one can use a combination of these methods [22], but these are three general approaches, with much academic research leveraging one or more of these methods [23–26]. See Table 1 for a summary of the advantages, disadvantages, and examples of implementations.

While site-centric web analytics tools, such as Google Analytics, can provide results for one's own website, there is often a need to compare with other websites, though Google Analytics does provide some limited benchmarking reports by industry (https://support.google.com/analytics/answer/6086666). Therefore, competitive benchmarking services, such as SimilarWeb, have become essential for web analytics in the business intelligence area [27]. These analysis services provide computational web analytics results for one or more websites, a critically needed capability for competitive research and analysis [28]. These website analytics services allow benchmarking of web analytics measures and metrics among multiple websites. Website analytics services are essential for a variety of reasons, including competitive analysis, advertising, marketing, domain purchasing, programmatic media buying [29–35], and firm acquisition [36], along with the use of website analytics services in academic research [37, 38]. They are also valuable for accessing the external view of one's own website (i.e., what others who do not have access to site-centric analytics data see). These website analytics services return a variety of metrics depending on the platform. However, there are questions concerning the accuracy and reliability of both types of analytics platforms, affecting billions of dollars in online advertising, firm acquisition, and research. As such, there is a critical need to assess these tools and the validity of the reported metrics.

In this research, we compare web analytics statistics from Google Analytics (the industry-standard website analytics platform at the time of the study) and SimilarWeb (the industry-standard traffic analytics platform at the time of the study) using four core web analytics metrics (i.e., *total visits*, *unique visitors*, *bounce rate*, and *average session duration*) averaged monthly over 12 months for 86 websites. We select SimilarWeb due to the scope of its data collection, reportedly one billion daily digital signals, two terabytes of daily analyzed data, more than two hundred data scientists employed, and more than ten thousand daily traffic reports generated, with reporting features better or as good than other services [39] at the time of the study. As such, SimilarWeb represents state-of-the-art in the online competitive analytics area. We leave the investigation of others services besides Google Analytics and SimilarWeb to other research. We conduct statistical analysis along several fronts reporting both exploratory

**Table 1. Comparison of user, site, and network-centric approaches to web analytics data collection showing advantages, disadvantages, and examples of each approach at the time of the study.**

| Approach | Advantages | Disadvantages | Examples |
|---|---|---|---|
| User-centric | • Focus on people<br>• Compare across websites; so can use for business intelligence | • Creating a representative user panel is challenging<br>• User computer software must be installed | • Alexa<br>• ComScore |
| Site-Centric | • No special user software to install<br>• Wide range of analytics for a specific site | • Site software must be installed<br>• Focus on cookies and tags, not real people<br>• Access is limited to the website owner; cannot use for business intelligence among multiple sites | • Google Analytics<br>• Adobe Analytics<br>• IBM Analytics |
| Network-Centric | • Data collection is straightforward<br>• No special software to install for users or sites<br>• Compare across websites; can use for business intelligence | • Data can be challenging to obtain<br>• Limited onsite analytics; generally only between sites data | • Hitwise<br>• SEMRush<br>• SpyFu |

and statistical results. We then tease apart the nuanced differences in the metrics and possible sources of error [40] and present the theoretical and the practical implications of this research. The techniques employed by Google Analytics are similar to techniques employed by other analytics platforms, such as Adobe Analytics, IBM Analytics, and Piwik Analytics. The techniques used by SimilarWeb are similar to the techniques of other website analytics services, such as Alexa, comScore, SEMRush, Ahrefs, and Hitwise, in the employment of user, site, and/or network data collection. So, the results of this research apply to a wide range of analytics tasks, most notably in the website domain, providing an enhanced understanding of the data underlying competitive intelligence and the use of such analytics platforms. Moreover, the metrics reviewed are commonly used in many industries employing online analytics, such as advertising, online content creation, and e-commerce. Therefore, the findings are impactful for several domains.

## Review of literature

Web analytics services have been employed in research and used by researchers for an array of inquiries and topics. These areas include, among others, online gaming [41], social media and multi-channel online marketing [42, 43], online community shopping [44], online purchase predictions [45, 46], online research methods [47], social science [48, 49], and user-generated content on social media [50–54]. These services have also been used in research concerning online interests in specific topics [55–57]–including online branding in social media [58, 59], online purchasing [60], and mobile application usage [61]. They have also been used in studies about website trust and privacy [62–66], website design [37, 67–69], and website popularity and ranking [42, 44, 70–77]–for a variety of areas. These prior studies indicate that analytics tools are widely used in peer-reviewed academic research and relied on for various metrics. However, to our knowledge, none of the prior research studies examined the accuracy of these website analytics services prior to employment.

Academic research on this area of analytics evaluation is limited. Lo and Sedhain [78] evaluate six websites lists, including the ranked list from Alexa (the only service employed in the study that is still active, as of the date of this research). The researchers examined the top 100 websites and compared the rankings among the lists. They concluded that the ranking among the lists differed. This difference is not surprising given that the methodologies used to create the study lists varied in terms of website traffic, number of backlinks, and opinions of human judges. Vaughan and Yang [79] use organizations from the United States (U.S.) and China and collect web traffic data for these sites from Alexa Internet, Google Trends for Websites, and Compete (Alexa is the only service still active from the study, as of the date of this research). The researchers did not evaluate the traffic services but instead reported correlations between web traffic data and measures of academic quality for universities. In a ComScore study, Napoli, Lavrakas, and Callegaro [80] present some of the challenges and issues with the user-centric analytics approach, namely that the results often do not align with site-centric measures. The researchers attribute the discrepancies to the sampling of the user panels. Scheitle and fellow researchers [19] examine several websites' rankings, including Alexa but not SimilarWeb, investigating similarity, stability, representativeness, responsiveness, and benignness in the cybersecurity domain, but they do not report actual analytics numbers. The researchers report that the ranked lists are unstable and open to manipulation. Pochat and colleagues [18] extend this research by introducing a list that is less susceptible to rank manipulation.

While few academic studies have examined analytics services, fewer have evaluated the actual analytics numbers; instead, they focus on the more easily accessible (and usually free) ranked lists. Studies are even rarer still on the performance of SimilarWeb, despite its standing

and reputation as an industry leader. Scheitle and colleagues [19] attribute this absence to SimilarWeb charging for its service, although the researchers do not investigate this conjecture. Regardless of the reason, the only academic study that we are aware of as of the date of this research that explicitly examines traffic numbers, including SimilarWeb, is Prantl and Prantl [24]. This study compares rankings among Alexa, SimilarWeb, and NetMonitor [80] for a set of websites in the Czech Republic, using NetMonitor as the baseline. The research only reports the traffic comparison between SimilarWeb and NetMonitor. The researchers, unfortunately, provide neither detailed exploratory analysis nor statistical analysis of the analytics comparison. Also, NetMonitor uses a combination of site and user-centric measures, so it is unclear how the traffic metrics are calculated. The researchers [24] report that Similar-Web over reports traffic compared to NetMonitor. They also note that SimilarWeb traffic results are +/- 30% compared to NetMonitor traffic measurements for 49% of the 487 websites.

Several industry reports have also compared site analytics, usually using Google Analytics, with the analytics reported by other services. Some of these reports [81–83] show website analytics services, notably SimilarWeb, reportedly underestimating traffic, as much as 30% to 50%, while other reports (84–88) claim SimilarWeb overestimates traffic, from 11% for large websites to nearly 90% for small ones [84]. SimilarWeb itself states that reported values among analytics services will vary +/- 20%. However, a trend is that SimilarWeb [85, 86] consistently ranks as the best or one of the best analytics services in the industry [87, 88], as noted by several industry practitioners [30, 32, 33, 89–91]. SimilarWeb consistently outperforms other services [92], with reported performance better sometimes in the double digits [93]. Even when the reported analytics numbers are off, the SimilarWeb results usually correlate with the baseline site traffic trends. The correlation is also positive relative to overall accuracy among sites [93].

Although providing insights into the area, there are potential issues regarding relying on industry reports, including possible questions on data appropriateness, lack of explicitly defined methods of analysis, and conflicts of interest (as some of these studies are performed by potential competitors of SimilarWeb). Also, some of these studies employ a small number of data points [81, 82, 94], making statistical analysis challenging. Other studies have a short temporal span [83, 88, 93], as there can be significant traffic fluctuations for sites depending on the time of year, or mainly high-traffic websites [83], which are easier to calculate. Finally, some reports have imprecise metric reporting [83, 84, 92, 95], raising doubt on the results, or a limited set of metrics [81, 83, 95] not central to analytics insights. Because of these potential issues, there is a critical need for a rigorous academic analysis of website analytics services to supplement these industry reports.

Given the substantial use of analytics services in academic research and their widespread use in the practitioner communities, there is a notable lack of research examining the accuracy of these services. Determining their accuracy is critical, given the extensive reliance on analytics numbers across many domains of research and practice [96]. However, due to the absence of academic studies in the area, several unanswered questions remain, including: *How accurate are these analytics services*? *How do they compare with other analytics methods*? *Are these analytics tools better (or worse) at measuring specific analytics metrics than other methods*? *Are the reported metrics valid*? These are essential questions that need addressing for critical evaluation of research findings and business decisions that rely on these services. Although the questions are conceptually straightforward, they are surprisingly difficult to evaluate in practice. This difficulty, especially in terms of data collection, may be a compounding factor for the dearth of academic research in the area.

### Research questions

Our research objective is to *compare and contrast the reported analytics measurements between SimilarWeb and Google Analytics* in support of the broader goal of comparing these two approaches for measuring analytics and evaluating their accuracy. To investigate this research objective, we focus on four core web analytics metrics–*total visits*, *unique visitors*, *bounce rate*, and *average session duration*–which we define in the methods section. Although there is a lengthy list of possible metrics for investigation, these four metrics are central to addressing online behavioral user measurements, including *frequency*, *reach*, *engagement*, and *duration*, respectively. We acknowledge that there may be some conceptual overlap among these metrics. For example, bounce rates are sessions with an indeterminate duration that may indicate a lack of engagement, but average session duration also provides insights into user engagement. Nevertheless, these four metrics are central to the web analytics analysis of nearly any single website or set of websites; therefore, they are worthy of investigation. In the interest of space and impact of findings, we focus on these four metrics, leaving other metrics for future research.

Given that Google Analytics uses site-centric website data and SimilarWeb employs a triangulation of datasets and techniques, we would reasonably expect values would differ between the two. However, is it currently unknown how much they differ, which is most accurate, or if the results are correlated. Therefore, because Google Analytics is, at the time of the study, the *de facto* industry standard for websites, we use Google Analytics measurements as the baseline for this research. Our hypotheses (H) are:

- **H1**: SimilarWeb measurement of *total visits* to websites differ from those reported by Google Analytics.

- **H2**: SimilarWeb measurement of *unique visitors* to websites differ from those reported by Google Analytics.

- **H3**: SimilarWeb measurement of *bounce rates* for websites differ from those reported by Google Analytics.

- **H4**: SimilarWeb measurement of *average session durations* for websites differ from those reported by Google Analytics.

We investigate these hypotheses using the following methodology.

## Material and methods

Our data collection platforms are Google Analytics and SimilarWeb. Each service is explained in the following subsections.

### Google analytics

Google Analytics is a site-centric web analytics platform and, at the time of the study, is the most popular site analytics tool in use [97]–that is, it is the market leader. Google Analytics tracks and reports website analytics for a specific website. This tracking by Google Analytics is accomplished via cookies and tags [98]; a tag is a snippet of JavaScript code added to the individual pages. The tags are executed in the JavaScript-enabled browsers of the website visitors. Once executed, the tag sends the visit data to a data server and sets a first-party cookie on cookie-enabled browsers on visitors' computers. The tag must be on a page on the site for Google Analytics to track the web analytics data for that page.

Concerning the data collection, analysis, and reporting algorithms of Google Analytics, they are proprietary. However, enough is known to validate their employment as being industry standard and state-of-the-art. The techniques of cookies and the general process of tagging are well-known, although there may be some nuances in implementation. Google Analytics employs statistical data sampling techniques [99], so the values in these cases may not be the result of the complete data analysis for some reports. However, the general overview of the data sampling approach is presented in reasonable detail [29], and the described subsampling is an industry standard methodology [100].

## SimilarWeb

SimilarWeb [85, 86] is a service providing web analytics data for one or multiple websites. SimilarWeb uses a mix of user, site, and network-centric data collection approaches to triangulate data [39, 101], reportedly collecting and analyzing billions of data points per day [22]. SimilarWeb's philosophical approach is that each method has strengths and weaknesses, and the best practice is triangulating multiple algorithms and data sources [39], a respected approach in data collection and analysis.

Regarding the data collection, analysis, and reporting algorithms of SimilarWeb, they are proprietary, but again, enough is known to validate the general implementation as state-of-the-art. The SimilarWeb foundational principle of triangulating user, site, and network-centric data collection data [39, 101] is academically sound, with triangulating data and methods used and advocated widely by scholars [5, 102]. SimilarWeb data collection, analysis, and reporting methodology are outlined in reasonable detail [22], although, like Google Analytics, the proprietary specifics are not provided. However, from the ample documentation that is available [22, 39, 86, 103–105], the general approach is to collect data from three primary sources, which are: (a) a reportedly 400 million worldwide user panel [103] at the time of the study, (b) specific website analytics tracking [39], and (c) ISP and other traffic data [39]. These sources are supplemented with publicly available datasets (e.g., population statistics). Each of these datasets will overlap (i.e., the web analytics data from one collection method will also appear in one or both of the other collection methods). With the collected data augmented with publicly available data [39], SimilarWeb uses statistical techniques and ensemble machine learning approaches to generate web analytics results. These analytics can then be compared to the overlapped data to make algorithmic adjustments to the predictions. This is a more complex approach relative to Google Analytics; however, SimilarWeb's scope of multiple websites also requires a more complicated approach. In sum, the general techniques employed by Similar-Web are standard methodologies [101, 106, 107], academically sound, and industry standard state-of-the-art.

## Data collection procedure

For our analysis, we identify a set of websites with analytics by SimilarWeb and having their Google Analytics accessible by SimilarWeb [104, 108], thereby making their Google Analytics values available. If a website has a Google Analytics associated, SimilarWeb, using the paid version, offers the option of reporting either the SimilarWeb or the Google Analytics numbers for these websites. For this access, the website owner grants SimilarWeb access to the website's Google Analytics account, so the data pull is direct. We verified this process with a website not employed in the study, encountering no issues with either access or reported data. This feature allows us to compare the SimilarWeb and the Google Analytics numbers for our identified web analytics metrics of total visits, unique visitors, bounce rates, and average session duration.

We employ the Majestic Million [108] to identify our pool of possible websites. The Majestic Million list of websites is creative commons licensed and derives from Majestic's web crawler. The Majestic Million list ranks sites by the number of /24 IPv4-subnets linking to that site, used as a proxy for website popularity. Using this large, open-licensed, and readily available list as the seed listing, we started at the top, submitted the link to the SimilarWeb application program interface (API), and checked whether SimilarWeb provided analytics or if the website associated its Google Analytics to the SimilarWeb service. We included it as a candidate website for our research if it had both SimilarWeb and Google Analytics metrics. If not, the website was excluded. We then proceeded to the following website on the list and repeated the submission and verification process.

We continued these steps until we identified 91 websites. There were five websites where Google Analytics and SimilarWeb values differed by orders of magnitude. As there seemed to be no discernible patterns among these five websites upon our examination, we excluded them as outliers and reserved them as candidates for future study. This action left us with 86 websites for analysis. We concluded that this was more than satisfactory for our research, as the number is adequate for statistical analysis [109].

We have determined not to make the specific links publicly available for the privacy of the companies' websites and given that these web analytics comparisons are a paid business product of SimilarWeb. However, we outline our methodology in detail so that those interested can recreate our research. Also, we provide the web analytics and related data concerning the websites (excluding website name and website link) in S1 File.

## Data analysis

We employ paired t-tests for our analysis. The paired t-test compares two means from the same population to determine whether or not there is a statistical difference. As the paired t-test is for normally distributed populations, we conduct the Shapiro-Wilk test for visits, unique visits, bounce rate, and average session duration for both platforms to test for normality. As expected, the Shapiro-Wilk tests showed a significant departure from the normality for all variables. Therefore, we transformed our data to a normal distribution via the Box-Cox transformation [110] using the log-transformation function, log(variable). We then again conducted the Shapiro-Wilk test; the effect sizes of non-normality were very small, small, or medium, indicating the magnitude of the difference between the sample and normal distribution. Therefore, the data is successfully normalized for our purposes, though a bit of skewness exists, as the data is weighted toward the center of the analytics numbers using the log transformation, as shown for visits in Fig 1.

Despite the existing skewness, previous work shows that a method such as the paired t-test is robust in these cases [111, 112]. The transformation ensured that our statistical approach is

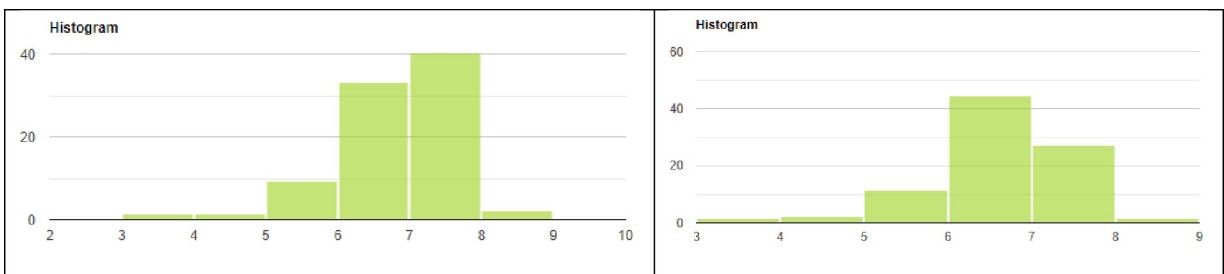

**Fig 1. Histogram of normalized Google Analytics and SimilarWeb visits data.** Effect sizes Are Very Small and Small Respectively, Indicating the Difference Between the Sample Distribution and the Normal Distribution is Very Small/Small.

valid for the dataset's distributions. We then execute the paired t-test on four groups to test the differences between the means of total visits, unique visitors, bounce rates, and average session duration on the transformed values.

Further, we employ the Pearson correlation test, which measures the strength of a linear relationship between two variables, using the normalized values for the metrics under evaluation. This correlation analysis informs us how the two analytics services rank the websites relative to each other for a given metric, regardless of the agreement on the absolute values. These analytics services are often employed in site rankings, which is a common task in many competitive intelligence endeavors and used in many industry verticals, so such correlation analysis is insightful for using the two services in various domains.

Using the SimilarWeb API, we collect the reported values for total visits, unique visitors, bounce rate, and average session duration for each month over 12 months (September 1, 2019, through August 31, 2020, inclusive) for each of the 86 websites on our list. We then average the monthly values for each metric for each platform to obtain the values that we use in our analysis. We use the monthly average to mitigate any specific monthly fluctuation. For example, some websites have seasonal fluctuations in analytics. Some websites may experience outages during specific months or denial of service attacks. Using the monthly average over 12 months helps mitigate the possible short-term variations.

Our four measures, total visits, unique visitors, bounce rate, and average session duration, are considered core metrics in the domain of web analytics [1, 113, 114]. A metric is typically a number, such as a count or a percentage. However, measuring or calculating these metrics may vary by platform or service; therefore, it is crucial to understand these differences. Additionally, the conceptual understanding of these metrics may differ from the specific ability of a method for tracking in implementation. Table 2 presents an overview of these metrics.

## Results

### Exploratory results

Our 86 websites represent companies based in 26 countries, as shown in Table 3. We used the country classifications provided by SimilarWeb, and we verified the classifications based on our assessment of the websites and links.

The 86 organizational websites are from the following 19 industry verticals, as shown in Table 4. We used the industry classifications provided by SimilarWeb [115, 116], and we verified the classifications based on our assessment of the websites and company background material provided.

The types of the 86 organizational websites are shown in Table 5. We used the site type classifications provided by SimilarWeb, and we verified the classification based on our assessment of the website content and features. Content sites are websites that provide content as their primary function. Transactional websites are sites that are primarily selling a product. 'Other' refers to those websites that do not fit into the other two categories.

### H1: Measurements of total visits differ

A paired t-test was conducted to compare the number of *total visits* reported by Google Analytics and SimilarWeb. There was a significant difference in the reported number of total visits for Google Analytics (M = 6.82, SD = 0.31) and SimilarWeb (M = 6.66, SD = 0.29); t(85) = 6.43, p < 0.01. These results indicate a difference in the number of total visits between the two approaches. Specifically, our results show that SimilarWeb's reported number of total visits is statistically <u>lower</u> than the values reported by Google Analytics. **Therefore, H1 is fully**

**Table 2. Comparison of definitions of total visits, unique visitors, bounce rate, and session duration conceptually and for the two analytics platforms: Google Analytics and SimilarWeb.**

| Definition of: | Total Visits | Unique Visitors | Bounced Rate | Average Session Duration |
|---|---|---|---|---|
| **Conceptually** | Sum of times that all people go to a website during a measurement period. A measure of frequency. | Sum of actual people who have visited a website at least once during a period. A measure of reach. | A bounced visit is the act of a person immediately leaving a website with no interaction. A measure of engagement. | The average length of time that visitors are on the website. A measure of duration. |
| **Practically** | Sum of times at least one page of a website has been loaded into a browser during a measurement period. | Sum of distinct tracking measures requesting pages from a website during a given period determined by a method such as cookie, tag, or plugin. | Ratio of single-page visits divided by all visits to a website during a given period (i.e., single page visits divided by all visits) | Total duration of all sessions divided by the number of sessions |
| **Google Analytics** | Sum of single visits to a website consisting of one or more pageviews during a measurement period. The default visit timeout is 30 minutes, meaning that if there is not activity for this visit on the website for more than 30 minutes, then a new visit will be reported if another interaction occurs. | Sum of unique Google Analytics tracking code and browser cookies that visit a website at least once during a measurement period. | Ratio of single-page visits divided by all visits to a website during a measurement period Single-page sessions have an undefined session duration, since there are no subsequent server hits after the first one that would let Analytics calculate the length of the session. However, using a period of inactivity for the exit, bounce sessions have a duration of zero. | Session duration is the period of a group of user interactions with a website from the first and subsequent interactions to a period of inactivity. By default, a session lasts until there are 30 minutes of inactivity. Session duration relies on a period of inactivity to end the session, as there is no server hit when the visitor exits the website. |
| **SimilarWeb** | Sum of times at least one page of a website has been loaded into a browser during a measurement period Subsequent page views are included in the same visit until the user is inactive for more than 30 minutes. If a user becomes active again after 30 minutes, that counts as a new visit. A new session will also start at midnight. | Sum of computing devices visiting a website within a geographical area and during a measurement period. | Ratio of single page visits by all visits for a website within a geographical area and during a measurement period. | Session duration is the period of is a group of user interactions with a website from the first and subsequent interactions to a period of inactivity. By default, a session lasts until there are 30 minutes of inactivity. |

**supported: SimilarWeb's measurements of total visits to websites differ from those reported by Google Analytics.**

The number of total visits for all 86 websites was 1,703.5 million (max = 292.5 million; min = 1,998, med = 7.8 million), as reported by Google Analytics, and 1,060.1 million (max = 140.8 million; min = 4,443; med = 5.9 million), as reported by SimilarWeb. Using the total aggregate visits for all 86 websites using Google Analytics as the baseline, SimilarWeb

**Table 3. Host country of organization for 86 websites in study.**

| Country | No. | % |
|---|---|---|
| United States | 43 | 50.0% |
| India | 6 | 7.0% |
| Russian Federation | 6 | 7.0% |
| Japan | 4 | 4.7% |
| United Kingdom | 4 | 4.7% |
| France | 3 | 3.5% |
| Israel | 3 | 3.5% |
| Spain | 2 | 2.3% |
| One each (Belarus, Belgium, Canada, Chile, China, Cuba, Ecuador, Germany, Madagascar, Malaysia, Nigeria, Taiwan, Turkey, Ukraine, United Arab Emirates) | 15 | 17.4% |
| | 86 | 100.0% |

**Table 4. Industry vertical of organization for 86 websites in study.**

| Website Category | No. | % |
|---|---|---|
| News and Media | 36 | 41.9% |
| Computers Electronics and Technology | 10 | 11.6% |
| Arts and Entertainment | 9 | 10.5% |
| Science and Education | 5 | 5.8% |
| Community and Society | 4 | 4.7% |
| Finance | 4 | 4.7% |
| Business and Consumer Services | 2 | 2.3% |
| E-commerce and Shopping | 2 | 2.3% |
| Gambling | 2 | 2.3% |
| Travel and Tourism/ | 2 | 2.3% |
| Vehicles | 2 | 2.3% |
| Health | 1 | 1.2% |
| Hobbies and Leisure | 1 | 1.2% |
| Home and Garden | 1 | 1.2% |
| Jobs and Career | 1 | 1.2% |
| Law and Government | 1 | 1.2% |
| Lifestyle/Beauty and Cosmetics | 1 | 1.2% |
| Lifestyle/Fashion and Apparel | 1 | 1.2% |
| Sports | 1 | 1.2% |
| | 86 | 100.0% |

underestimated by 643 million (19.4%) total visits. Using Google Analytics numbers as the baseline for total visits, SimilarWeb overestimated 15 (17.4%) sites and underestimated 66 (76.7%) sites. The two platforms were nearly similar (~+/- 5%) for 5 (5.8%) sites.

Ranking the websites by total visits based on Google Analytics and SimilarWeb, we then conduct a Pearson correlation coefficient test. There was a significant strong positive association between the ranking of Google Analytics and SimilarWeb, $rs(85) = .954$, $p < .001$.

Graphically, we compare the reported total visits between Google Analytics and Similar-Web in Fig 2, showing the correlational relationship. As shown in Fig 2, the number of total visits between Google Analytics and SimilarWeb has a strong, positive, linear correlation.

This finding implies that, although the reported total visits values differ between the two platforms, the *trend* for the set of websites is generally consistent. So, if one is interested in a ranking (e.g., "Where does website X rank within this set of websites based on total visits?"), then SimilarWeb values will generally align with those of Google Analytics for those websites. However, if one is specifically interested in numbers (e.g., "What is the number of total visits to each of N websites?), then the SimilarWeb total visit numbers will be ~20% below those reported by Google Analytics, on average.

**Table 5. Website type for the 86 websites in study.**

| Site Type | No. | % |
|---|---|---|
| Content | 50 | 58.1% |
| Other | 34 | 39.5% |
| Transactional | 2 | 2.3% |
| | 86 | 100.0% |

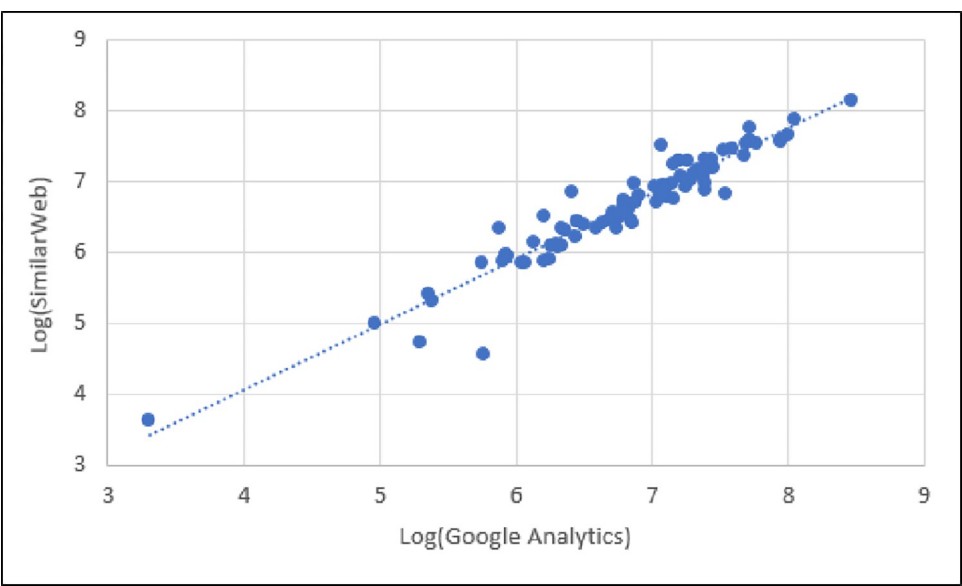

**Fig 2. Scatter plot of total visits reported by Google Analytics and SimilarWeb showing strong, positive, linear correlation.**

## H2: Measurements of unique visitors differ

A paired t-test was conducted to compare the number of *unique visitors* reported by Google Analytics and SimilarWeb. There was a significant difference in unique visitors for the Google Analytics (M = 6.56, SD = 0.26 million) and the SimilarWeb (M = 6.31, SD = 0.25) conditions; t(85) = 12.60, p < 0.01. These results indicate a difference in the number of unique visitors between the two approaches. Specifically, our results show that the reported number of unique visitors by SimilarWeb is statistically <u>lower</u> than the values reported by Google Analytics. **Therefore, H2 is fully supported: SimilarWeb measurement of unique visitors to websites differ from those reported by Google Analytics.**

The total number of unique visitors for all 86 websites was 834.7 million (max = 138.1 million; min = 1,799; med = 4.3 million) reported by Google Analytics and 439.0 million (max = 54.6 million; min = 2,361; med = 2.3 million) reported by SimilarWeb. Using the mean aggregate unique visitors for all 86 websites, using Google Analytics as the baseline, SimilarWeb underestimated by 395.6 million (38.7%) unique visitors. Using Google Analytics numbers as the baseline, SimilarWeb overestimated 4 (4.7%) sites and underestimated 82 (95.3%) sites.

Ranking the websites by unique visitors based on Google Analytics and SimilarWeb, we then conduct a Pearson correlation coefficient test. There was a significant strong positive association between the ranking of Google Analytics and SimilarWeb, rs(85) = .967, p < .001.

Graphically, we compare the reported unique visitors between Google Analytics and SimilarWeb in Fig 3, showing the correlational relationship. As shown in Fig 3, the number of total visits between Google Analytics and SimilarWeb has a strong, positive, linear correlation.

This finding indicates that, while the reported values for unique visitors differ between the two platforms, the *trend* for the set of websites is mostly consistent. So, if one is interested in a ranking (e.g., "Where does website X rank within this set of websites based on unique visitors?"), then SimilarWeb values will generally align with those of Google Analytics for those websites. However, if one is specifically interested in numbers (e.g., "What is the number of unique visitors to each of N websites?), then the SimilarWeb unique visitor numbers will be ~40% below those reported by Google Analytics, on average.

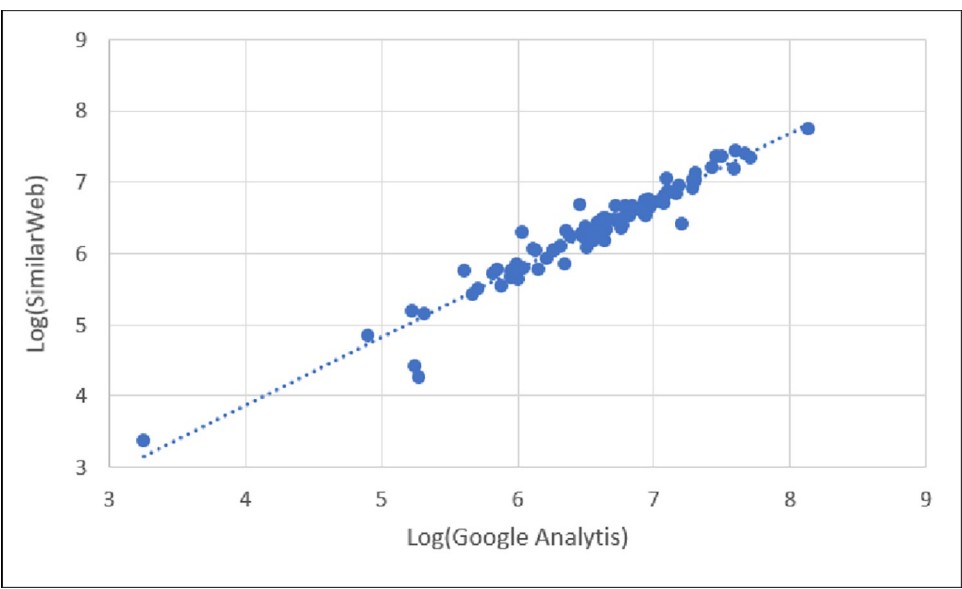

**Fig 3. Scatter plot of unique visitors reported by Google Analytics and SimilarWeb showing strong, positive, linear correlation.**

## H3: Measurements of bounce rates differ

A paired t-test was conducted to compare *bounce rates* reported by Google Analytics and SimilarWeb. There was a significant difference in the bounce rates between the Google Analytics (M = 0.58, SD = 0.03) and the SimilarWeb (M = 0.63, SD = 0.02) conditions; t(85) = -2,96, p < 0.01. Specifically, our results showed that the reported bounce rate by SimilarWeb was significantly <u>higher</u> than that reported by Google Analytics, **fully supporting H3: SimilarWeb measurement of bounce rates for websites differ from those reported by Google Analytics.**

The average of bounce rate for all 86 websites was 56.2% (SS = 20.4%, max = 88.9%; min = 20.4%; med = 59.2%) reported by Google Analytics and 63.0% (SS = 13.8%, max = 86.0%; min = 28.8%; med = 65.3%) as reported SimilarWeb. Using Google Analytics as the baseline, SimilarWeb analytics were 6.8% more than the average bounce rate. Additionally, SimilarWeb over calculated 35 (40.7%) sites and under calculated 31 (36.0%) sites. The two platforms were nearly similar (+/- 5) for 20 (23.3%) sites.

We then conducted a Pearson correlation coefficient test to rank the websites by bounce rate based on Google Analytics and SimilarWeb. There was a significant positive association between the ranking of Google Analytics and SimilarWeb, rs(85) = .461, p < .001.

Graphically, this is illustrated in Fig 4, where we compare bounce rates between Google Analytics and SimilarWeb. As shown in Fig 4, the bounce rates between Google Analytics and SimilarWeb have a moderate, positive, linear correlation.

This finding indicates that, although SimilarWeb and Google Analytics report similar bounce rates for more than 20% of the sites, the difference between the values for the other 80% for the two platforms was high. We address the possible reasons for this high discrepancy later in the discussion of results.

## H4: Measurements of average session duration differ

A paired t-test was conducted to compare the *average session duration* reported by Google Analytics and SimilarWeb. There was a significant difference in the average session duration

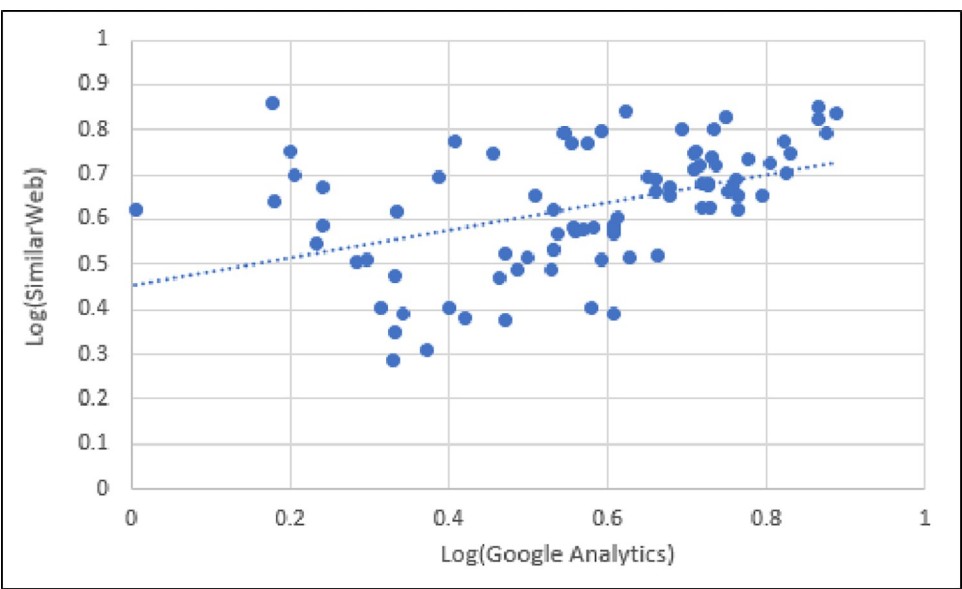

**Fig 4. Scatter plot of bounce rate reported by Google Analytics and SimilarWeb showing moderate, positive, linear correlation.**

between the Google Analytics (M = 2.15, SD = 0.05) and the SimilarWeb (M = 2.47, SD = 0.71) conditions; t(85) = -8.59, p < 0.01. Specifically, our results showed that the reported average session duration by SimilarWeb was significantly <u>higher</u> than that reported by Google Analytics, **fully supporting H4: SimilarWeb measurement of average session duration for websites differ from those reported by Google Analytics.**

The average session duration for all 86 websites was 202.91 seconds (SS = 239.71, max = 1439.51; min = 33.25; med = 119.63) reported by Google Analytics and 463.51 seconds (SS = 640.99, max = 4498.08; min = 62.42; med = 267.13) as reported SimilarWeb. Using Google Analytics as the baseline, SimilarWeb reported a 52.6% more total average session duration. Additionally, SimilarWeb over reported 63 (73.3%) sites and under reported 9 (10.5%) sites, relative to Google Analytics. The two platforms were nearly similar (~+/- 5) for 14 (16.3%) sites.

Ranking the websites by average session duration based on Google Analytics and SimilarWeb, we then conduct a Pearson correlation. There was a significant positive association between the ranking of Google Analytics and SimilarWeb, rs(85) = .536, p < .001, as shown in Fig 5.

This finding indicates that, although SimilarWeb and Google Analytics report similar average sessions for about 16% of the sites, the difference between the values for the other 84% of the sites for the two platforms was generally high. We address the possible reasons for this high discrepancy later in the discussion of results.

## Discussion

### General discussion

Table 6 summarizes our findings for the 86 websites using average monthly total visits, unique visitors, bounce rate, and average session duration during the 12-month analysis period.

As shown in Table 6, statistical testing of all four hypotheses is statistically significant, so **<u>all four hypotheses are supported</u>**. The reported values for total visits, unique visitors, bounce

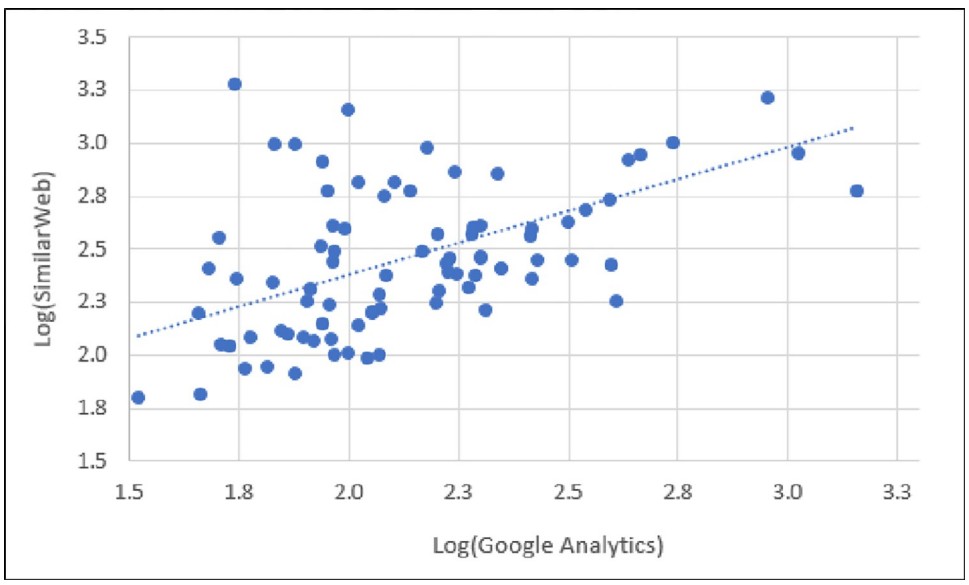

**Fig 5. Scatter plot of average session duration reported by Google Analytics and SimilarWeb showing moderate, positive, linear correlation.**

rates, and average session duration for Google Analytics and SimilarWeb differ significantly. The website rankings by each service are significantly correlated, so it seems that these ranked lists can be used for research on analytics, competitive analysis, and analytics calculations for a set of websites, with the caveat highlighted in [18, 19]. These analyses compare the two services' precision (i.e., how close measured values are to each other).

However, the underlying question motivating our research remains this: *How accurate are the reported metrics from website analytics services (i.e.*, how close are the reported values to the 'true' values)? Regardless of the statistical testing results, this motivational question is more challenging to address. In reality, there is one 'true' number of visits, visitors, bounces, and average session duration. However, is it realistic to expect any web analytics service to match reality perfectly? Moreover, what is the reality in terms of web analytics? In our perspective, it is a misconception to view web analytics data collection as "counting." In most cases, web analytics is not counting; instead, it is "measuring." It is well known that there will be an error rate

**Table 6. Summary of results comparing Google Analytics and SimilarWeb for total visits, unique visitors, bounce rate, and average session duration.** Difference uses Google Analytics as the Baseline. Results based on Paired t-Test for Hypotheses Supported.

| Metric / Service | Google Analytics | SimilarWeb | Difference | Hypotheses |
|---|---|---|---|---|
| Total Visits | 1,703,584,207 | 1,060,137,189 | 19.4% | Fully Supported–The reported values differ |
| Unique Visitors | 834,656,530 | 439,016,436 | 38.7% | Fully Supported–The reported values differ |
| Bounce Rate | 56.2% | 63.0% | 6.8% | Fully Supported–The reported values differ |
| Average Session Duration | 202.91 | 463.51 | 56.2% | Fully Supported–The reported values differ |
| Number of Sites (Relative to Google Analytics Values) Where SimilarWeb Numbers Were: | | | | |
| | Higher | Lower | Similar (~+/- 5%) | |
| Total Visits | 15 (17.4%) | 66 (76.7%) | 5 (5.8%) | SimilarWeb values will generally be <u>lower</u> than Google Analytics |
| Unique Visitors | 4 (4.7%) | 82 (95.3%) | 0 (0.0%) | SimilarWeb values will generally be <u>lower</u> than Google Analytics |
| Bounce Rate | 35 (40.7%) | 31 (36.0%) | 20 (23.3%) | SimilarWeb values will generally be <u>higher</u> than Google Analytics |
| Average Session Duration | 63 (73.3%) | 9 (10.5%) | 14 (16.3%) | SimilarWeb values will generally be <u>higher</u> than Google Analytics |

(+/- n%) for nearly any measure [117]. No measure or measurement tool is perfect, and web data can be particularly messy.

Although one might lean toward considering metrics reported by Google Analytics as the 'gold standard' for website analytics (and justifiably so in many cases), it is also known within the industry that Google Analytics has tracking issues in some cases. Also, a reportedly high percentage of Google Analytics accounts are incorrectly set up [118–121], perhaps skewing the measuring in some cases. There are also contexts where other analytics methods might be more appropriate. Google Analytics relies on one data collection approach: basically, a cookie and tagging technique. There are certainly cases (e.g., cleared cookies, incognito browsing) when this method is inaccurate (e.g., unique visitors). Furthermore, Google Analytics might have different settings in terms of filtering, such as housekeeping visits from organizational employees that would slant the results. Therefore, these concerns result in issues with Google Analytics being seen as the 'gold standard.'

To investigate our motivation research question regarding the accuracy of Google Analytics and SimilarWeb as analytics services, we conduct a deductive analysis using a likelihood of error [122]. We analyze what makes theoretical sense for which web analytics approach, Google Analytics or SimilarWeb, would result in the most accurate measurement for each of our metrics. We discuss our analysis of each metric below.

**Bounce rate (engagement).**   A high bounce rate is undesirable for many sites. The bounce rate means that someone comes to a site and leaves without taking relevant action. For this metric, **both** Google Analytics [123] and SimilarWeb are conceptually incorrect due to the practical issues of measuring a bounce visit [124]. For a meaningful session measurement, there must be an entry point (where the person came to the site) and an exit point (where the person left the site). If there is no endpoint to the session, both Google Analytics and SimilarWeb count it as a single page visit and a bounce because there is no exit interaction.

There are many situations where relevant action is taken on a site, but there is no exit point [125]. For example, there can be an e-commerce site where a potential consumer arrives on a product page, reads the content, and takes no other action at that time. Another case is a newspaper site where an audience member comes to the site, scans the headlines, reads the article snippets, but takes no other action, such as clicking [126]. In each of these cases, the visit could last several minutes or longer. However, since there is no exit page (i.e., no second page), Google Analytics and SimilarWeb would count these example visits as bounces.

So, we can reasonably assume both Google Analytics and SimilarWeb are overcounting bounces, conceptually. This may be why the values vary substantially between the two services. However, since bounce is a site-centric specific measure, we would expect Google Analytics to be more precise (if not more accurate) than SimilarWeb when measuring bounce rate on a single given site. However, SimilarWeb's panel data may help correct this somewhat for a set of sites, which Google Analytics does not measure. So, if one needs to examine the bounce rate of several websites, Google Analytics cannot be used since website owners usually do not make their web analytics data available to the public.

In terms of mechanical metrics, one would expect Google Analytics to be better for an individual site. SimilarWeb might be expected to give reasonable bounce rate numbers for some sites due to their user-centric panel data, and bounce rates are generally high, especially for highly trafficked sites. This reasonableness in results from both Google Analytics and SimilarWeb is borne out in our statistical analysis above, where the two services were more in agreement for the larger traffic sites for bounce rates (see Fig 4).

**Average session duration (duration).**   Again, for this metric, **both** Google Analytics [123] and SimilarWeb are conceptually incorrect due to the practical issues of measuring the end of a session. Similar to the bounce rate, there is no exit point (i.e., where the person left the site).

As there is no endpoint, both Google Analytics and SimilarWeb rely on a temporal timeout measured from the time of the last interaction. So, this most likely under measures the duration of many sessions.

Again, since average session duration is a site-centric specific measure, Google Analytics would be expected to be at least more precise (if not more accurate) than SimilarWeb when measuring average session duration on a single given site. Again, SimilarWeb's panel data may somewhat help correct this for a set of sites for which Google Analytics data is unavailable. So, similar to bounce rate, if one needs to examine the average session duration of several websites, Google Analytics cannot be used, as this data is usually not public. In the end, conceptually, both Google Analytics and SimilarWeb are most likely under measuring average session duration. In terms of practical implementation, one would expect Google Analytics to be better for an individual site. SimilarWeb might be expected to give reasonable numbers for some sites due to their user-centric panel data.

**Total visits (frequency).**    This seems like a straightforward site-centric metric for which Google Analytics should excel. Although there is room for some noise in the visits, such as housekeeping visits (i.e., visits from internal company personnel for site maintenance), bot-generated visits [127], purchased traffic, or hacking attacks that might not conceptually meet the definition of a visit, it is difficult to imagine how an analytics service could be better than a site-centric service in this regard. Using the site and network-centric data collection data employed by website analytics services like SimilarWeb would not mitigate some of the noise mentioned above; however, the user-centric panel data might compensate for some of the noise issues for at least a high traffic website and for bot traffic. However, in general, one would expect Google Analytics to be more accurate in measuring visits than SimilarWeb. However, Google Analytics data is generally unavailable for multiple websites, so relying on Google Analytics is not practical for these situations. For these cases, one would need to employ an analytics service, such as SimilarWeb. Based on our analysis above, values for total visits from SimilarWeb would be less than Google Analytics measurements by ~20% on average.

**Unique visitors (reach).**    Finally, we consider unique visitors. In this case, perhaps surprisingly, one would expect the greater likelihood of error to be with the site-centric measurements, resulting in SimilarWeb measures being more accurate.

Site-centric services, such as Google Analytics, typically rely on a combination of cookies and tags to measure unique visitors. This approach would generally result in an overcount of unique visitors by the service. For example, the expected life cycle of a computer is three to five years [128, 129], meaning a person changing computers would be registered as a new visitor. The market share of browsers has changed considerably over the years [130, 131], meaning when someone has changed browsers, he/she would be registered as a new visitor. Studies show that 40% of Internet users clear cookies daily, weekly, or monthly [132, 133], and about 3.7% of users disable all cookies [134, 135]. These actions would trigger a unique visitor count when visiting a website. Some studies point to a much higher rate, with more than 30% of users deleting cookies in a given month [132]. Many people also use the incognito mode on their browsers [136, 137], triggering a new visitor count in Google Analytics [138, 139]. Also, many people have multiple devices (e.g., personal computer, work computer, smartphone, tablet), with about 50% of Americans, for example, using four Internet-enabled devices [140, 141], so each device would be counted as a unique visitor even if it is the same person is using the multiple devices.

For these reasons, the unique visitor number measured using the cookie approach would likely lead to an overcount using site-centric metrics. *How much of an overcount*? Based on the issues just outlined, it seems that, for Google Analytics, a 20% overestimate in monthly unique

visitors to 30% overestimate for more extended periods seems reasonable. However, more precise measures require an in-depth study and are a task for future research.

For unique visitors, it seems that panel data, such as those that Similar Web and other network-centric services use, might be more accurate. However, this might only hold for larger websites. It is not clear that panel data would be accurate for lower-traffic websites as there is not enough traffic to these sites to generate reasonable statistical analysis. Generally, for unique visitors, it seems that Google Analytics would most likely overestimate the number of unique visitors to the website. SimilarWeb might be more accurate for the larger traffic websites due to its user panel data approach but have questionable accuracy (either over- or underestimating) for the smaller traffic websites. Again, this conclusion is borne out by our analysis above, where the difference between Google Analytics and SimilarWeb increased for the smaller websites (see Fig 3).

## Theoretical implications

We highlight three theoretical implications of this research, which are:

- **Triangulation of Data, Methods, and Services**: There seems, at present, to be no single data collection approach (user, site, or network-centric) or web analytics service (including Google Analytics or SimilarWeb) that would be effective for all metrics, contexts, or business needs. Therefore, a triangulation of services, depending on the data, method of analysis, or need, seems to be the most appropriate approach. It appears reasonable that user-centric approaches can be leveraged for in-depth investigation of user online behaviors, albeit usually with a sample. Site-centric approaches can be leveraged for the investigation of users' onsite behaviors. Network-centric approaches can be leveraged for in-depth investigation of user intersite behaviors (i.e., navigation between sites).

- **Discrepancies with Implementation**: Regarding precision, we have established differences between the two services, and we know the general methodologies and metrics calculations. However, the nuances of implementation have not been independently audited as of the date of this study. So, in practice, we cannot say definitely which is the best implementation for a given metric. Again, this points to the need for triangulation of methods. It also highlights the lack of a gold standard for evaluating website analytics services. Regardless of any nuances in implementation, the values between the two services are correlated, and, as discussed above, we can infer the preferred approach using deductive analysis.

- **Discrepancies with Reality**: Precision does not mean accuracy for either Google Analytics or SimilarWeb. We have already outlined potential issues with all four of the metrics examined (i.e., total visits, unique visitors, bounce rates, average session duration). The application mechanics are not aligned with conceptual definitions of what these metrics supposedly measure. This situation calls for both continued research into improved measures and a realization that the reported values (from both Google Analytics and SimilarWeb) are not counts per se and should not be viewed necessarily as 'truth.' Rather, the values are reported measures with some error rates (+/-).

## Practical implications

We highlight three practical implications of the findings, which are:

- **Use of Google Analytics and SimilarWeb**: Findings of our research show that, in general, SimilarWeb results for total visits and number of unique visitors will generally be lower than those reported by Google Analytics, and the correlation between the two platforms is high

for these two metrics. So, if one is interested in ranking a set of websites for which one does not have the Google Analytics data, the SimilarWeb metrics are a workable proxy. If one is interested in the actual Google Analytics traffic for a set of websites, one can use the Similar-Web results and increase by about 20% for total visits and about 40% for unique visitors, on average. As a caveat, the Google Analytics unique visitor's numbers are probably an over-count, and the SimilarWeb values may be more in line with reality. As an easier 'rule of thumb', we suggest using a 20% adjustment (i.e., increase SimilarWeb numbers) for both metrics based on the analysis findings above. The realization that these services can be complementary can improve decision-making that relies on KPIs and metrics from website analytics data.

- **Verification of Analytics for a Single Website**: In general, Google Analytics is a site-centric web analytics platform, so it would be a reasonable service to use for a single website that one owns and has access. However, comparing analytics values from Google Analytics to those of SimilarWeb (or other website analytics services) may be worthwhile, as these will be the values that outsiders see concerning the website.

- **Estimating Google Analytics Metrics for Multiple Websites**: As shown above, the differences between Google Analytics and SimilarWeb metrics for total visits and unique visitors are systematic (i.e., the differences stay relatively constant), notably for visits and unique visitors. This means, if you have Google Analytics values for one site, you can adjust and use a similar difference for the other websites to get reasonable analytics numbers to those from Google Analytics. This technique is valuable in competitive analysis situations where you compare multiple sites against a known website and want the Google Analytics values for all sites. However, SimilarWeb generally provides conservative analytics metrics compared to Google Analytics, meaning that, if solely relying on this single service, analytics measures may be lower, especially for onsite interactions. So, decisions using these analytics metrics need to include this as a factor.

## Limitations, future research, and strengths

**Limitations and future research.**   The first limitation concerns data quality. In the absence of ground truth, we primarily measure precision and not the accuracy of the two web analytics services. As noted, there are inconsistencies between the two platforms. So, the analytics data that decision-makers may perceive as accurate, objective, and correct may not have these qualities due to the several potential sources for errors outlined above. Web analytics services should undertake future research to provide metric values with confidence intervals to depict them as ranges rather than exact values. Another limitation is that the source codes and specific implementations for either of these platforms are not available, so the nuances of the implementations cannot be verified. Although it is apparent from results and from company materials that both platforms use state-of-the-art algorithmic approaches, future research could focus on using open-source analytics platforms, such as Matomo [142], to tease apart some of these metric implementations. An additional limitation is that a large percentage of the sites used in this research are content creation sites based in the U.S.A., which might skew user behavior. Other future research involves replication studies with different sets of websites, other website analytics services, other metrics, and analysis of specific website segments based on type, size, industry vertical, or country (i.e., China being a critical region of interest).

**Strengths.**   There are several strengths of this research. First, we use two popular web analytics services. Second, we employ 86 websites with various attributes, ensuring a robust sample size. Third, we collect data over an extended period of 12 months to mitigate for short

periods of fluctuation with the website analytics measures. Fourth, we report and statistically evaluate four core web analytics metrics–total visits, unique visitors, bounce rates, and average session duration. Fifth, we discuss and offer theoretical and practical implications of our research. To our knowledge, this is one of the first and one of the most extensive academic examinations and analyses of these popular web analytics services.

## Conclusion

For this research, we compared four analytics metrics from Google Analytics to those from SimilarWeb based on 12 months of data for 86 diverse websites. Findings show statistically significant differences between the two services for total visits, unique visitors, bounce rates, and average session duration. Compared to Google Analytics, SimilarWeb values were ~20% *lower* for total visits, ~40% *lower* for unique visitors, ~25% *higher* for bounce rate, and ~50% *higher* for average session duration, on average. The rankings of all four metrics are significantly correlated between Google Analytics and SimilarWeb, and the measurement differences are systematic between the two analytics services. The implications are that SimilarWeb provides conservative analytics results relative to Google Analytics, and these web analytics tools can be complementarily utilized in various contexts, especially when having data for one website and needing analytics data for other websites.

## Supporting information

**S1 File.**
(DOCX)

## Author Contributions

**Conceptualization:** Bernard J. Jansen, Joni Salminen.

**Formal analysis:** Bernard J. Jansen.

**Investigation:** Bernard J. Jansen.

**Methodology:** Bernard J. Jansen.

**Writing – original draft:** Bernard J. Jansen.

**Writing – review & editing:** Bernard J. Jansen, Soon-gyo Jung, Joni Salminen.

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
