## [Decision Letter · Decision Letter 0]

13 Jan 2022

PONE-D-21-03616

Measuring website interaction: A comparison of two industry standard analytic approaches using 86 websites

PLOS ONE

Dear Dr. Jansen,

Thank you for submitting your manuscript to PLOS ONE. After careful consideration, we feel that it has merit but does not fully meet PLOS ONE’s publication criteria as it currently stands. Therefore, we invite you to submit a revised version of the manuscript that addresses the points raised during the review process.

The reviewers agree that there is clear merit in this work, with some positive statements to this effect.  However, some reviewer comments are requests for clarification and this needs to be addressed.  In particular, reviewers have commented on the metrics used, their appropriateness and how they are used and interpreted.  There are also numerous comments about the statistical analysis that require a response, and/or clarification and/or update in the artlcle.  Reviewers also converge on requesting more reflection and discussion of results and implications of the work.

We look forward to receiving your revised manuscript.

Kind regards,

Hussein Suleman, PhD

Academic Editor

PLOS ONE

Journal Requirements:

2. Please note that in order to use the direct billing option the corresponding author must be affiliated with the chosen institute. Please either amend your manuscript to change the affiliation or corresponding author, or email us at plosone@plos.org with a request to remove this option.

3. We note that Figures 1 and 2 in your submission contain copyrighted images. All PLOS content is published under the Creative Commons Attribution License (CC BY 4.0), which means that the manuscript, images, and Supporting Information files will be freely available online, and any third party is permitted to access, download, copy, distribute, and use these materials in any way, even commercially, with proper attribution. For more information, see our copyright guidelines: http://journals.plos.org/plosone/s/licenses-and-copyright.

a. You may seek permission from the original copyright holder of Figures 1 and 2 to publish the content specifically under the CC BY 4.0 license. 

Reviewers' comments:

Reviewer's Responses to Questions

**Comments to the Author**

1. Is the manuscript technically sound, and do the data support the conclusions?

Reviewer #1: Yes

Reviewer #2: Yes

Reviewer #3: Yes

2. Has the statistical analysis been performed appropriately and rigorously? 

Reviewer #1: No

Reviewer #2: Yes

Reviewer #3: Yes

3. Have the authors made all data underlying the findings in their manuscript fully available?

Reviewer #1: Yes

Reviewer #2: Yes

Reviewer #3: Yes

4. Is the manuscript presented in an intelligible fashion and written in standard English?

Reviewer #1: Yes

Reviewer #2: Yes

Reviewer #3: Yes

5. Review Comments to the Author

Reviewer #1: General comment: The authors had made a meritorious effort and tried effectively to compare the produced values of two analytics platforms. The literature review that has been used is in quite good relationship with the research problematic. Moreover, an important effort has been made to address the practical contribution of the paper to other researchers or practitioners. However, there are some major issues. First the selection of bounce rate metric is not thoroughly aligned with the meaning of duration (see further justification in the comments 8-9 below). Second, there is some difficult understanding regarding the statistical tests that have been selected and what was finally presented in results (See comments 11-15). Lastly, both theoretical and practical contributions are unfolded in an organized and logical way. But it will be very useful to add even more practical contributions from a competitive intelligence point of view. What this manuscript offers compared to the prior relative research approaches on the field of Web Analytics validity and competitive intelligence strategy? And how the up-to-date theoretical scientific approaches are benefited from this paper? Once more, well done for your effort, and I hope the forthcoming suggestions/comments will help you to optimize the value of this paper.

1. Line 32. we need to be more explicit here. What other uses are available based on the citation 2?

2. Table 1. Line 76-77. In the third column the Ahrefs tool is more a backlink checking tool and not a behavioural analytics platform. Better not to include it and refer some other tool more relevant with the web behavioural analytics and not with the off-site optimization and backlinks building.

3. Lines 81-83. The same thing is conducted also with SEMrush as well. And more specifically, SEMRush provides explicit statistics on a daily basis for competitors through graphs, figures etc. And also, the triangulation perspective is adopted on SEMrush, just like SimilarWeb. So why we choose SimilarWeb compared to the others? In a general sense, it will reinforce furtherly the justification of the paper if we put a clear paragraph or a table referring that compared to the others, we choose SimilarWeb for these reasons (one, two, three, four, six, ten and so on reasons).

4. Line 101. ok this is good! But for what reason? the generalization of results to a wide range of analytics technologies what gives to the practical and research community? For example, greater competitive intelligence strategy? Better WA platforms design and capabilities? Something like that.

5. Line 102. Hmm, there are other metrics within these platforms. Mostly user-centric (average video duration, avg videos watched in channel, different types of engagement with a post, followers/subscribers gain and so on). Hence, the results of this study cannot impact on several other domain, but only between web analytics platforms that estimate only websites traffic. Better not to include this assumption.

6. Line 122-123. this seems to be a little be general sentence about their findings. What these correlations specifically depict? And actually, I suppose, that the purpose here in our paper is to present prior works that focus on the comparison of web traffic platforms, to find differences and fluctuations among them. Not to compare web traffic stats with organizational performance. So, it needs to be more explicit here.

7. Line 133-134. Please guys, reform this sentence. Personally, I believe that this is a little bit arrogant, and does not express academic ethos. It just like that saying "ok you there Scheitle and colleagues, you don't have money, but we have money, and we can do research and you cannot ;) . Probably it is true, but better to redefine this sentence.

8. Line 179. hmm ok Frequency is related with total visits per a determined time-range, Reach is related with the unique visitors. But duration is related mostly with visit duration and page per visit as metrics. Bounce rate express the immediate abandonment from a website without proceeding to any kind of interaction with the content thus this mean zero duration. Probably we can assume here that bounce rate is related mostly with content usability and representativeness of users search terms with what they retrieved as websites’ content from search engine results. That is, if we do not have a good alignment of search term and content, then we have high bounce rate and vice-versa. Or if we have poor usability, then bounce rate is increased as well. So better change the duration with something else more specific that is aligned in a better way with the bounce rate. In a general sense, the involvement of bounce rate metric and its inclusion under the meaning of measuring duration is one of the main issues within the paper. The metric itself is a little bit vexed and you pointed this out in your argumentation including several related references. In continuation of this comment, I try to help you more with another one comment related with bounce rate included in Table 2.

9. Table 2. Column 3. we mentioned “A bounced visit is the act of a person immediately leaving a website before any interaction can reasonably occur” This point is conflicting with the below one point "measure of duration". Bounce rate is not a measure of duration, so if there is no interaction, there is no duration. And based on Google as you stated below within the table << Bounce rate is single-page sessions divided by all sessions, or the percentage of all sessions on your site in which users viewed only a single page and triggered only a single request to the Analytics server. These single-page sessions have a session duration of 0 seconds since there are no subsequent hits after the first one that would let Analytics calculate the length of the session. >> Probably you take it from here. at: https://support.google.com/analytics/answer/1009409?hl=en#:~:text=Bounce%20rate%20is%20single%2Dpage,request%20to%20the%20Analytics%20server. Therefore, I am afraid that we cannot use Bounce rate within the whole paper. And I do not understand why we do not use pages per session or time spent as metrics for measuring duration. This also measures the depth of exploration.

10. Line 238. Reading the citation (number 107) and the paper itself from the acm, it is a little bit fuzzy how large-scale machine learning on a social media such as twitter is related with the SimilarWeb standard methods as it is mostly a website traffic intelligence tool and not a social media competitive intel platform.

11. Line 244-245. How confident we are that this linking process extracts the specific analytics from google analytics without deviations from the original one source, namely the GA platform? Ok till now, we are sure that the provided GA data within the similar web platform have differentiations with the provided similar web data. Ok very good on that. But are we sure that GA data within the SimilarWeb are the same with the original data extracted from GA platform for the examined websites? In other words, do we proceed into a preliminary comparison at the same time-period between the extracted google analytics data from the two platforms, that is original GA and GA data as they included within SimilarWeb? Or can we ask the admins of these Google Analytics Accounts if they can ensure -even in a small sample of the websites (5 or 10 of the total 86)- that the provided Google Analytics data from the Google Analytics platform are the same with the provided final Google Analytics data from the Similar Web? This for sure, will overhaul the trustworthiness of our research sample and also the validity of our methodology.

12. Line 255-256. Well, we do not agree into this assumption guys. Who says that the rule of thumb is about 30 websites and not 31 or 29 for descriptives? Better reinforce it with a citation here. You can retrieve it even from a statistical perspective paper (such as the citation 109 that you have already used). Or from the prior approaches that related with web analytics platforms comparisons and their gathered sample compared to ours in this paper. As it is now, is more an opinion, and not a documented justification.

13. Line 267-274. 1) Ok, if someone search, based on literature review, we need normal distribution to execute paired t-test. Now based on our implications here in this paragraph, we do not have normal distribution at the initial dataset. And indeed, after downloading the file from the Supporting Information, we discover very high skewness values within the items. Also we extract a low value of Shapiro-Wilk which has been conducted for testing normality and linearity of the sample.

1)Therefore, we need first a non-parametric test to prove that our data are not normally distributed or in other words to prove that all the variables do not follow normal distribution (we can also prove it with Wilcoxon signed rank test, the Mann-Whitney U Test and the Kruskal-Wallis test).

2) After proving non-normality then we take on the Box-Cox transformation. And ok this is good lads, as we deployed it. After that, we argue here that we have normal distribution even there is a bit of skewness. But which is the normality value of the variables right now after the transformation? This is missing. So here we need to re-run a second test to prove that we transformed our data and now we are in the right order; we have the required normality to conduct paired t-test. Therefore, we need to conduct a normal distribution and stating that the results indicating that after the data has been transformed, we have a normal distribution.

3) After the transformation of the data through box-cox how they shaped? how they transformed. What numbers where existing previously and how they are shaped now after the transformation. It will be useful to provide a small sample (4-5 websites in the three variables) within a table on how the dataset was; and how the dataset has been transformed right now after the box-cox.

4) Thereafter, our method to adopt paired t-test will be furtherly reinforced by the citations you included (111 and 112)

14. Line 276-277. Hmm might be a little bit confusing for the reader. So, we conducted the tests on the transformed data. That is good. But we report the non-transformed values? Why this choice lads? Why we conducted the transformation? Probably to make the dataset normal-distributed. But we present the non-transformed values? Therefore, so why to conduct the transformation before? And actually, the non-transformed values without the transformation would give a greater clarity as we refer here. Sorry guys for not understanding this choice, but we need to be more explicit for the sake of the forthcoming readers. Thank you.

15. Line 323-325. Oh guys hold on a second. Here the Spearman coefficient comes from the sky, without mentioned nothing within the Methodology section about its scope and what will give to the readers. We have mentioned on the theoretical part some things about correlations, but reading and reading again the theory, I cannot understand what this correlation will practically gives to us. How we interpret it? that is why we correlate them? And why we use Spearman instead of Pearson? Secondly, spearman is deployed mostly on non-normally distributed datasets. Have we conducted the Sprearman on the non-transformed dataset or on the transformed one? If it is the latter, then it needs Pearson which is conducted mostly on normal distributions.

In any case, if there is a reason for conducting Correlation Analysis then we must:

A) Refer with clarity why we do this and what proves in support to the scope of the paper.

B) Refer clearly in which dataset you have applied the Correlation Analysis. It is the non-transformed or transformed one. If it is the latter one, then Pearson is more appropriate.

C) Include scatter plots for all the three correlations for the involved metrics. the high numbers of coefficients say almost nothing to a demanding reader.

16. Regarding figures 6-8. They need improvement. What these numbers mean both in vertical and horizontal axes? And especially on the horizontal one. Although the comparison through the line is comprehensible, the rest are not. Also, we can minimize the white space (where it is possible) by eliminating the range of the vertical access.

17. Line 405. We refer “that these ranked lists can be used for research and other purposes”. Ok but for what other purposes? this is a little be general. Better to be more explicit here and point out the other purposes.

18. Line 414. This citation (118) is related with the messy situation in Scientometrics and has nothing to do with the web analytics of websites. Better find something else, or remove it.

19. Line 420. there is no "installed correctly or the same on all the websites". The script is one. If it is installed, then produces numbers. If it is not, then no numbers. Of course, there are incorrections between the connections of GA with Google Ads, Search Console or other platforms and their produced metrics. But in case of the three metrics that have been used here, there are measured properly or indicated zero values if there is a problem in set up. In addition, if we have doubts about the proper installation of GA, why we do not use the Tag Assistant Legacy of Google as browser extension in our data collection section? This tool identifies errors in analytics installation (check here https://chrome.google.com/webstore/detail/tag-assistant-legacy-by-g/kejbdjndbnbjgmefkgdddjlbokphdefk?hl=en)

20. Line 433-438. Again, regarding the Bounce Rate metric. Well, this is a contradictory justification with the aforementioned definition of Bounce Rate as can be seen in table 2. And if we want to consider the Duration as the third central measurement of Web Analytics, why we choose bounce rate which is at least contentious for many cases in the literature review regarding duration validity? And not choosing the visit duration (SW) and the Avg. Visit Duration (GA) to make a comparison among them? This will eliminate all these doubts about bounce rate validity.

21. Line 525. These two citations on this line. The first one 119, refers issues about the setup errors of GA. However, none of these errors of administrators affect the three metrics that we involve here. For example, if we were involving demographics, then ok, we have validity problems. But none of the statements of Alex Ramadan affect the Total visits, Unique Visitors and Bounce Rate. The other link (citation 119) is broken as a 404 page.

22. Regarding reference list. Citations 28, 32, 33, 54, 55, 84 are broken or are not working properly.

End of Comments/Suggestions.

Thank you for this opportunity.

Reviewer #2: This is a very well-written manuscript. Very easy to read. The material is well-organized.

The manuscript deals with an important problem area: the accuracy of popular website analytics and traffic estimation services (e.g., Google Analytics and SimilarWeb). The manuscript identifies and addresses a research gap: a lack of academic research and interest in studying web analytics.

To improve, can the authors provide more insight on why there is a lack of attention among academics in currently studying this phenomenon? In the abstract, rather than saying the accuracy of metrics provided by Google Analytics and SimilarWeb will be discussed, provide a short sentence or two that speaks to or describes the accuracy of these metrics. In the paper, provide more insight on what the impact of SimilarWeb providing conservative traffic metrics compared to Google Analytics actually means in terms of practice. Why should we care? Why is it important to know that SimilarWeb and Google Analytics can be used in a complementary fashion when direct website data is not available? How important is this to know? Elaborate more on the implications of this research.

There are 143 references included in this paper. This is great, but over the top. I think the references can be reduced to a more significant subset. This would reduced the paper's word count.

What impact do the study's findings have on user-centric, site-centric, and network-centric approaches to web analytics data collection identified earlier in the paper?

The USA represents half of the 86 websites studied. News and media content represents 42% of the 86 websites. It would be good to further describe how the this skewed sample affects the findings and interpretation of results.

Reviewer #3: The authors conducted a comparison between Google Analytics and SimilarWeb based on analytics metrics data. The results provide both theoretical and practical implications. The paper is clearly organized and well written. With some minor improvements this piece is worth publishing, and I have a few specific suggestions.

First, the authors need to justify their selection of total visits, unique visitors, and bounce rates as the three metrics. Why excluding other common metrics such as time on site/page?

Second, all three hypotheses are supported, but how does this help evaluate the accuracy of the two analytics services? I think it impossible to indicate which one is more accurate given the significant differences between them in terms of the three metrics.

Finally, I suggest that the authors improve their discussion section by providing more insights into the causes for the differences between Google Analytics and SimilarWeb.

A minor problem - I’m confused by the statement “The techniques used by SimilarWeb are similar to the techniques of other traffic services, such as Alexa, comScore, SEMRush, Ahrefs, and Hitwise.” (Page 5, Line 100). While Alexa and comScore user-centric, SEMRush, Ahrefs, and Hitwise are network-centric. Why “similar”? What are the “techniques”?

6. PLOS authors have the option to publish the peer review history of their article (what does this mean?). If published, this will include your full peer review and any attached files.

Reviewer #1: **Yes: **Georgios A. Giannakopoulos and Ioannis C. Drivas

Reviewer #2: No

Reviewer #3: No

---

## [Author Response · Author response to Decision Letter 0]

28 Feb 2022

PONE-D-21-03616: Measuring user interactions with websites: A comparison of two industry standard analytics approaches using data of 86 websites

This response letter contains our replies to the reviewers’ suggestions and comments. We include our comments in italics. We reference locations in the manuscript that specifically address major suggestions. Where appropriate, we provide a snippet from the manuscript. In this version of the manuscript, we also highlight the many significant changes made from the prior submission. 

We believe we have addressed both the spirit and the specifics of the reviewers’ comments in the manuscript’s current version. 

We thank the reviewers for their many detailed and constructive comments that have greatly improved the research presented in this version of the manuscript. 

META-REVIEW

Thank you for submitting your manuscript to PLOS ONE. After careful consideration, we feel that it has merit but does not fully meet PLOS ONE’s publication criteria as it currently stands. Therefore, we invite you to submit a revised version of the manuscript that addresses the points raised during the review process.

We thank the editor for this opportunity to revise our manuscript. 

With this new round of comments, we believe we have addressed both the spirit and the specifics of the reviewers’ comments in the manuscript’s current version, as outlined below. 

Thanks! 

The reviewers agree that there is clear merit in this work, with some positive statements to this effect.

We thank the reviewers for their positive comments about the research presented in this manuscript, as we also believe that the research has clear merit. 

However, some reviewer comments are requests for clarification and this needs to be addressed. In particular, reviewers have commented on the metrics used, their appropriateness and how they are used and interpreted. There are also numerous comments about the statistical analysis that require a response, and/or clarification and/or update in the artlcle. Reviewers also converge on requesting more reflection and discussion of results and implications of the work.

We believe we have addressed both the spirit and the specifics of the reviewers’ comments in the manuscript’s current version, as outlined below. 

Again, thanks! 

Please submit your revised manuscript by Feb 26 2022 11:59PM. 

Thank you for the information. We are submitting the revised manuscript within the required timeframe. Actually, we are early! 

We look forward to receiving your revised manuscript.

Again, thanks! We hope that you like it! 

Comments to the Author

1. Is the manuscript technically sound, and do the data support the conclusions?

Reviewer #1: Yes

Reviewer #2: Yes

Reviewer #3: Yes

We thank the reviewers for their positive support of the research presented in this manuscript, as we also believe that the research is technically sound. 

2. Has the statistical analysis been performed appropriately and rigorously?

Reviewer #1: No

Reviewer #2: Yes

Reviewer #3: Yes

We thank the reviewers for their positive support of the research presented in this manuscript, as we also believe that the statistical analysis has been performed appropriately and rigorously (R#2 and R#3). Concerning revisions, we believe we have addressed both the spirit and the specifics of the reviewers’ (R#1) suggestions in the manuscript’s current version

3. Have the authors made all data underlying the findings in their manuscript fully available?

Reviewer #1: Yes

Reviewer #2: Yes

Reviewer #3: Yes

We thank the reviewers for their positive support of the research presented in this manuscript, as we make all data underlying the findings described in this manuscript fully available. 

4. Is the manuscript presented in an intelligible fashion and written in standard English?

Reviewer #1: Yes

Reviewer #2: Yes

Reviewer #3: Yes

We thank the reviewers for their positive support of the research presented in this manuscript. 

-------------

5. Review Comments to the Author

Reviewer #1: General comment: The authors had made a meritorious effort and tried effectively to compare the produced values of two analytics platforms. 

We thank the reviewer for the positive comment about the manuscript, which we also believe is a meritorious effort to effectively compare the produced values of two analytics platforms. 

The literature review that has been used is in quite good relationship with the research problematic. 

We thank the reviewer for the positive comment about the manuscript, as we also believe that the literature review used is in a quite good relationship with the research. 

Moreover, an important effort has been made to address the practical contribution of the paper to other researchers or practitioners. 

We thank the reviewer for the positive comment about the manuscript, as we also believe that an important effort has been made to address the practical contribution of the paper to other researchers or practitioners. 

However, there are some major issues. First the selection of bounce rate metric is not thoroughly aligned with the meaning of duration (see further justification in the comments 8-9 below). 

We thank the reviewer for pointing this out. Great catch! Upon reflection, you are correct. We have pivoted from the use of bounce rate as a measure of duration in this version of the manuscript, which we discuss in detail below. 

Second, there is some difficult understanding regarding the statistical tests that have been selected and what was finally presented in results (See comments 11-15).

We thank the reviewer for the suggestions on better presenting the statistical tests that have been selected and what was finally presented in the results. We address the suggestions in this manuscript version, which we discuss in detail below. 

Lastly, both theoretical and practical contributions are unfolded in an organized and logical way. But it will be very useful to add even more practical contributions from a competitive intelligence point of view. What this manuscript offers compared to the prior relative research approaches on the field of Web Analytics validity and competitive intelligence strategy? And how the up-to-date theoretical scientific approaches are benefited from this paper? 

We thank the reviewer for these suggestions on better presenting the practical and theoretical contributions. We address the suggestions in this version of the manuscript, which we discuss in detail below. 

Once more, well done for your effort, and I hope the forthcoming suggestions/comments will help you to optimize the value of this paper.

Thanks so much for the ‘well done’! We put a lot of work, spirit, thought, and heart into this research and manuscript, and the same for incorporating and implementing your suggestions! They improved the manuscript! 

We thank you so much for the excellent suggestions and comments. Some made us think and pivot our direction. Others helped clarify, and others strengthened the research. We sincerely appreciate the thoroughness and thoughtfulness of the review. 

With this round of revisions, we believe we have addressed both the spirit and the specifics of the reviewers’ comments in the manuscript’s current version as outlined below. The manuscript and reporting of the research are better because of your suggestions.

Again, thanks! 

1. Line 32. we need to be more explicit here. What other uses are available based on the citation 2?

We thank the reviewer for this suggestion which we now address in this version of the manuscript.

Page 2, 35: Web analytics is a critical component of business intelligence, competitive analysis, website benchmarking, online advertising, online marketing, and digital marketing (2) as business decisions are made based on website traffic measures obtained from website analytics services. 

2. Table 1. Line 76-77. In the third column the Ahrefs tool is more a backlink checking tool and not a behavioural analytics platform. Better not to include it and refer some other tool more relevant with the web behavioural analytics and not with the off-site optimization and backlinks building.

We thank the reviewer for this suggestion which we now address in this version of the manuscript. We removed the mention of Ahrefs.

See Table 1, page 4

3. Lines 81-83. The same thing is conducted also with SEMrush as well. And more specifically, SEMRush provides explicit statistics on a daily basis for competitors through graphs, figures etc. And also, the triangulation perspective is adopted on SEMrush, just like SimilarWeb. So why we choose SimilarWeb compared to the others? In a general sense, it will reinforce furtherly the justification of the paper if we put a clear paragraph or a table referring that compared to the others, we choose SimilarWeb for these reasons (one, two, three, four, six, ten and so on reasons).

We thank the reviewer for this suggestion which we now address in this version of the manuscript.

Page 5, 96: In this research, we compare web analytics statistics from Google Analytics (the industry-standard website analytics platform at the time of the study) and SimilarWeb (the industry-standard traffic analytics platform at the time of the study) using four core web analytics metrics (i.e., total visits, unique visitors, bounce rate, and average session duration) averaged monthly over 12 months for 86 websites. We select SimilarWeb due to the scope of its data collection, reportedly one billion daily digital signals, two terabytes of daily analyzed data, more than two hundred data scientists employed, and more than ten thousand daily traffic reports generated, with reporting features better or as good than other services (39) at the time of the study. As such, SimilarWeb represents state-of-the-art in the online competitive analytics area. We leave the investigation of others services besides Google Analytics and SimilarWeb to other research.

5. Line 102. Hmm, there are other metrics within these platforms. Mostly user-centric (average video duration, avg videos watched in channel, different types of engagement with a post, followers/subscribers gain and so on). Hence, the results of this study cannot impact on several other domain, but only between web analytics platforms that estimate only websites traffic. Better not to include this assumption.

We thank the reviewer for this suggestion which we now address in this version of the manuscript.

Page 6, 112: Moreover, the metrics reviewed are commonly used in many industries employing online analytics, such as advertising, online content creation, and e-commerce. Therefore, the findings are impactful for several domains. 

6. Line 122-123. this seems to be a little be general sentence about their findings. What these correlations specifically depict? And actually, I suppose, that the purpose here in our paper is to present prior works that focus on the comparison of web traffic platforms, to find differences and fluctuations among them. Not to compare web traffic stats with organizational performance. So, it needs to be more explicit here.

We thank the reviewer for this pointing this out, and we now address this clarification in this version of the manuscript.

Page 6, 133: The researchers did not evaluate the traffic services but instead reported correlations between web traffic data and measures of academic quality for universities. 

7. Line 133-134. Please guys, reform this sentence. Personally, I believe that this is a little bit arrogant, and does not express academic ethos. It just like that saying “ok you there Scheitle and colleagues, you don’t have money, but we have money, and we can do research and you cannot ;) . Probably it is true, but better to redefine this sentence.

To clarify, this was implied in the cited paper, not by us. However, we take your point and reword the sentence in this version of the manuscript. 

Page 7, 144: Scheitle and colleagues (19) attribute this absence to SimilarWeb charging for its service, although the researchers do not investigate this conjecture.

8. Line 179. hmm ok Frequency is related with total visits per a determined time-range, Reach is related with the unique visitors. But duration is related mostly with visit duration and page per visit as metrics. Bounce rate express the immediate abandonment from a website without proceeding to any kind of interaction with the content thus this mean zero duration. Probably we can assume here that bounce rate is related mostly with content usability and representativeness of users search terms with what they retrieved as websites’ content from search engine results. That is, if we do not have a good alignment of search term and content, then we have high bounce rate and vice-versa. Or if we have poor usability, then bounce rate is increased as well. So better change the duration with something else more specific that is aligned in a better way with the bounce rate. In a general sense, the involvement of bounce rate metric and its inclusion under the meaning of measuring duration is one of the main issues within the paper. The metric itself is a little bit vexed and you pointed this out in your argumentation including several related references. In continuation of this comment, I try to help you more with another one comment related with bounce rate included in Table 2.

Again, we thank the reviewer for pointing this out. Really helpful! Upon reflection, you are correct, and we pivoted from the use of bounce rate as a measure of duration in this version of the manuscript. 

Page 8, 187: To investigate this research objective, we focus on four core web analytics metrics – total visits, unique visitors, bounce rate, and average session duration – which we define in the methods section. Although there is a lengthy list of possible metrics for investigation, these four metrics are central to addressing online behavioral user measurements, including frequency, reach, engagement, and duration, respectively. 

9. Table 2. Column 3. we mentioned “A bounced visit is the act of a person immediately leaving a website before any interaction can reasonably occur” This point is conflicting with the below one point “measure of duration”. Bounce rate is not a measure of duration, so if there is no interaction, there is no duration. And based on Google as you stated below within the table << Bounce rate is single-page sessions divided by all sessions, or the percentage of all sessions on your site in which users viewed only a single page and triggered only a single request to the Analytics server. These single-page sessions have a session duration of 0 seconds since there are no subsequent hits after the first one that would let Analytics calculate the length of the session. >> Probably you take it from here. at: https://support.google.com/analytics/answer/1009409?hl=en#:~:text=Bounce%20rate%20is%20single%2Dpage,request%20to%20the%20Analytics%20server. Therefore, I am afraid that we cannot use Bounce rate within the whole paper. And I do not understand why we do not use pages per session or time spent as metrics for measuring duration. This also measures the depth of exploration.

Again, we thank the reviewer for pointing this out. Really helpful! We really need to keep the reporting of the bounce rate, as it is a very common metric for website analytics. However, in this version of the manuscript, we re-focus bounce rate as a measure of engagement rather than duration.

Additionally, we now have added an additional section for analysis of average session duration for the duration metric. 

See Table 2, page 14

10. Line 238. Reading the citation (number 107) and the paper itself from the acm, it is a little bit fuzzy how large-scale machine learning on a social media such as twitter is related with the SimilarWeb standard methods as it is mostly a website traffic intelligence tool and not a social media competitive intel platform.

We thank the reviewer for pointing out the possible irrelevant reference, which we have removed in this version of the manuscript. 

Specifically, we removed: Brownlee J. A Tour of Machine Learning Algorithms [Internet]. Machine Learning Mastery. 2019 [cited 2020 Oct 6]. Available from: https://machinelearningmastery.com/a-tour-of-machine-learning-algorithms/

Page 12, 252: In sum, the general techniques employed by SimilarWeb are standard methodologies (101,106,107), academically sound, and industry standard state-of-the-art. 

11. Line 244-245. How confident we are that this linking process extracts the specific analytics from google analytics without deviations from the original one source, namely the GA platform? Ok till now, we are sure that the provided GA data within the similar web platform have differentiations with the provided similar web data. Ok very good on that. But are we sure that GA data within the SimilarWeb are the same with the original data extracted from GA platform for the examined websites? In other words, do we proceed into a preliminary comparison at the same time-period between the extracted google analytics data from the two platforms, that is original GA and GA data as they included within SimilarWeb? Or can we ask the admins of these Google Analytics Accounts if they can ensure -even in a small sample of the websites (5 or 10 of the total 86)- that the provided Google Analytics data from the Google Analytics platform are the same with the provided final Google Analytics data from the Similar Web? This for sure, will overhaul the trustworthiness of our research sample and also the validity of our methodology.

Concerning the process of acquiring the Google Analytics data, it is a straightforward access to Google, so the data is being pulled directly from the connected Google Analytics account. We now mention this in the report, and our verification of the data access. 

Page 12, 258: For this access, the website owner grants SimilarWeb access to the website’s Google Analytics account, so the data pull is direct. We verified this process with a website not employed in the study, encountering no issues with either access or reported data.

12. Line 255-256. Well, we do not agree into this assumption guys. Who says that the rule of thumb is about 30 websites and not 31 or 29 for descriptives? Better reinforce it with a citation here. You can retrieve it even from a statistical perspective paper (such as the citation 109 that you have already used). Or from the prior approaches that related with web analytics platforms comparisons and their gathered sample compared to ours in this paper. As it is now, is more an opinion, and not a documented justification.

We thank the reviewer for pointing this out concerning the exact number thirty. We address this comment by removing the offending sentence from this version of the manuscript. 

13. Line 267-274. 1) Ok, if someone search, based on literature review, we need normal distribution to execute paired t-test. Now based on our implications here in this paragraph, we do not have normal distribution at the initial dataset. And indeed, after downloading the file from the Supporting Information, we discover very high skewness values within the items. Also we extract a low value of Shapiro-Wilk which has been conducted for testing normality and linearity of the sample.

1)Therefore, we need first a non-parametric test to prove that our data are not normally distributed or in other words to prove that all the variables do not follow normal distribution (we can also prove it with Wilcoxon signed rank test, the Mann-Whitney U Test and the Kruskal-Wallis test).

2) After proving non-normality then we take on the Box-Cox transformation. And ok this is good lads, as we deployed it. After that, we argue here that we have normal distribution even there is a bit of skewness. But which is the normality value of the variables right now after the transformation? This is missing. So here we need to re-run a second test to prove that we transformed our data and now we are in the right order; we have the required normality to conduct paired t-test. Therefore, we need to conduct a normal distribution and stating that the results indicating that after the data has been transformed, we have a normal distribution.

3) After the transformation of the data through box-cox how they shaped? how they transformed. What numbers where existing previously and how they are shaped now after the transformation. It will be useful to provide a small sample (4-5 websites in the three variables) within a table on how the dataset was; and how the dataset has been transformed right now after the box-cox.

4) Thereafter, our method to adopt paired t-test will be furtherly reinforced by the citations you included (111 and 112)

The comment from the reviewer (“And ok this is good lads, …), made us smile! �

We thank the reviewer for these suggestions, which we now address in this version of the manuscript.

For (1), we conduct the Shapiro-Wilk test for each variable and platform. In the interest of space and the general expectation by most readers that the data will not be normal, we do not include the results in the manuscript. However, we provide them below to show that we did conduct them. 

• Google Analytics Visits: The Shapiro-Wilk test showed a significant departure from the normality, W(86) = .486, p < .001

• Google Analytics Unique Visitors: The Shapiro-Wilk test showed a significant departure from the normality, W(86) = .497, p < .001

• Google Analytics Bounce Rate: The Shapiro-Wilk test showed a significant departure from the normality, W(86) = .955, p = .004

• Google Analytics Session Duration: The Shapiro-Wilk test showed a significance departure from the normality, W(86) = .586, p < .001

• SimilarWeb Visits: The Shapiro-Wilk test showed a significant departure from the normality, W(86) = .584, p < .001

• SimilarWeb Unique Visitors: The Shapiro-Wilk test showed a significant departure from the normality, W(86) = .592, p < .001

• SimilarWeb Bounce Rate: The Shapiro-Wilk test showed a significant departure from the normality, W(86) = .967, p = .026

• SimilarWeb Session Duration: The Shapiro-Wilk test showed a significance departure from the normality, W(86) = .536, p < .001

For (2) and (3), we now included a graph (for total visits) of the transformed data in the manuscript after transformation. We conducted the Shapiro-Wilk test for each variable and platform post transformation. In the interests of space, we do not include the full results in the manuscript but do mention the effect sizes in the manuscript. We provide the effect size below to show that we did conduct them. We also include in this version of the manuscript, as suggested, a histogram of the distribution for one variable (total visits) as an example for the readers.

• Google Analytics Visits: The observed effect size KS - D is very small, 0.06478. This indicates that the magnitude of the difference between the sample distribution and the normal distribution is very small.

• SimilarWeb Visits: The observed effect size KS - D is small, 0.09128. This indicates that the magnitude of the difference between the sample distribution and the normal distribution is small.

• Google Analytics Unique Visits: The observed effect size KS - D is medium, 0.1065. This indicates that the magnitude of the difference between the sample distribution and the normal distribution is medium.

• SimilarWeb Unique Visits: The observed effect size KS - D is medium, 0.1065. This indicates that the magnitude of the difference between the sample distribution and the normal distribution is medium.

• Google Analytics Bounce Rate: The observed effect size KS - D is small, 0.09476. This indicates that the magnitude of the difference between the sample distribution and the normal distribution is small.

• SimilarWeb Bounce Rate: The observed effect size KS - D is small, 0.08994. This indicates that the magnitude of the difference between the sample distribution and the normal distribution is small. 

• Google Analytics Bounce Rate: The observed effect size KS - D is medium, 0.1066. This indicates that the magnitude of the difference between the sample distribution and the normal distribution is medium.

• SimilarWeb Bounce Rate: The observed effect size KS - D is small, 0.09481. This indicates that the magnitude of the difference between the sample distribution and the normal distribution is small.

For (4), as the reviewer stated, our position for using the paired t-test is now further supported by the mentioned references. Thanks for the suggestions! 

Again, we thank the reviewer for these suggestions concerning providing support for our analysis.

Page 13: We conducted the Shapiro-Wilk test for visits, unique visits, and bounce rate for both platforms. The Shapiro-Wilk tests showed a significant departure from the normality for all variables. 

See Figure 3, page 13. 

14. Line 276-277. Hmm might be a little bit confusing for the reader. So, we conducted the tests on the transformed data. That is good. But we report the non-transformed values? Why this choice lads? Why we conducted the transformation? Probably to make the dataset normal-distributed. But we present the non-transformed values? Therefore, so why to conduct the transformation before? And actually, the non-transformed values without the transformation would give a greater clarity as we refer here. Sorry guys for not understanding this choice, but we need to be more explicit for the sake of the forthcoming readers. Thank you.

We thought the actual values would be of more impact to the readers. However, it is a stylistic point. We take your comment that, if you wanted the transformed data than other readers may want the transformed data also. We now report the transformed data in this version of the manuscript. 

Page 13, 281: We employ paired t-tests for our analysis. The paired t-test compares two means from the same population to determine whether or not there is a statistical difference. As the paired t-test is for normally distributed populations, we conduct the Shapiro-Wilk test for visits, unique visits, bounce rate, and average session duration for both platforms to test for normality. As expected, the Shapiro-Wilk tests showed a significant departure from the normality for all variables. Therefore, we transformed our data to a normal distribution via the Box-Cox transformation (110) using the log-transformation function, log(variable). We then again conducted the Shapiro-Wilk test; the effect sizes of non-normality were very small, small, or medium, indicating the magnitude of the difference between the sample and normal distribution. Therefore, the data is successfully normalized for our purposes, though a bit of skewness exists, as the data is weighted toward the center of the analytics numbers using the log transformation, as shown for visits in Figure 3. 

Figure 3: Histogram of Normalized Google Analytics and SimilarWeb Visits Data. Effect sizes Are Very Small and Small Respectively, Indicating the Difference Between the Sample Distribution and the Normal Distribution is Very Small/Small

Despite the existing skewness, previous work shows that a method such as the paired t-test is robust in these cases (111,112). The transformation ensured that our statistical approach is valid for the dataset’s distributions. We then execute the paired t-test on four groups to test the differences between the means of total visits, unique visitors, bounce rates, and average session duration on the transformed values. 

15. Line 323-325. Oh guys hold on a second. Here the Spearman coefficient comes from the sky, without mentioned nothing within the Methodology section about its scope and what will give to the readers. We have mentioned on the theoretical part some things about correlations, but reading and reading again the theory, I cannot understand what this correlation will practically gives to us. How we interpret it? that is why we correlate them? And why we use Spearman instead of Pearson? Secondly, spearman is deployed mostly on non-normally distributed datasets. Have we conducted the Sprearman on the non-transformed dataset or on the transformed one? If it is the latter, then it needs Pearson which is conducted mostly on normal distributions.

In any case, if there is a reason for conducting Correlation Analysis then we must:

A) Refer with clarity why we do this and what proves in support to the scope of the paper.

B) Refer clearly in which dataset you have applied the Correlation Analysis. It is the non-transformed or transformed one. If it is the latter one, then Pearson is more appropriate.

C) Include scatter plots for all the three correlations for the involved metrics. the high numbers of coefficients say almost nothing to a demanding reader.

We thank the reviewer for this suggestion which we now address in this version of the manuscript. Specifically, we now discuss the correlation analysis in the methods section of the manuscript, including the use of the analysis. As we use the normalized versions of the data, we report the results of the Pearson correlations in this version of the manuscript. We highlight several places in the manuscript where we discuss correlations between the two analytics platforms. We address the scatter plots in the next comment of this Response to the Reviewers. 

Abstract: The website rankings between SimilarWeb and Google Analytics for all metrics are significantly correlated, especially for total visits and unique visitors. 

Page 9, 196: Given that Google Analytics uses site-centric website data and SimilarWeb employs a triangulation of datasets and techniques, we would reasonably expect values would differ between the two. However, is it currently unknown how much they differ, which is most accurate, or if the results are correlated. Therefore, because Google Analytics is the de facto industry standard for websites, we use Google Analytics measures as the baseline for this research.

Page 14, 298: Further, we employ the Pearson correlation test, which measures the strength of a linear relationship between two variables, using the normalized values for the metrics under evaluation. This correlation analysis informs us how the two analytics services rank the websites relative to each other for a given metric, regardless of the agreement on the absolute values. These analytics services are often employed in site rankings, which is a common task in many competitive intelligence endeavors and used in many industry verticals, so such correlation analysis is insightful for using the two services in various domains.

Page 17, 349: Ranking the websites by total visits based on Google Analytics and SimilarWeb, we then conduct a Pearson correlation coefficient test. There was a significant strong positive association between the ranking of Google Analytics and SimilarWeb, rs(85) = .954, p < .001. 

Page 18, 358: This finding implies that, although the reported total visits values differ between the two platforms, the trend for the set of websites is generally consistent. So, if one is interested in a ranking (e.g., “Where does website X rank within this set of websites based on total visits?”), then SimilarWeb values will generally align with those of Google Analytics for those websites. However, if one is specifically interested in numbers (e.g., “What is the number of total visits to each of N websites?), then the SimilarWeb total visit numbers will be ~20% below those reported by Google Analytics, on average. 

Page 19, 377: Ranking the websites by unique visitors based on Google Analytics and SimilarWeb, we then conduct a Pearson correlation coefficient test. There was a significant strong positive association between the ranking of Google Analytics and SimilarWeb, rs(85) = .967, p < .001. 

Page 20, 404: We then conducted a Pearson correlation coefficient test to rank the websites by bounce rate based on Google Analytics and SimilarWeb. There was a significant positive association between the ranking of Google Analytics and SimilarWeb, rs(85) = .461, p < .001. 

Page 21, 428: Ranking the websites by average session duration based on Google Analytics and SimilarWeb, we then conduct a Pearson correlation. There was a significant positive association between the ranking of Google Analytics and SimilarWeb, rs(85) = .536, p < .001.

Page 27, 579: Use of Google Analytics and SimilarWeb: Findings of our research show that, in general, SimilarWeb results for total visits and number of unique visitors will generally be lower than those reported by Google Analytics, and the correlation between the two platforms is high for these two metrics. So, if one is interested in ranking a set of websites for which one does not have the Google Analytics data, the SimilarWeb metrics are a workable proxy. If one is interested in the actual Google Analytics traffic for a set of websites, one can use the SimilarWeb results and increase by about 20% for total visits and about 40% for unique visitors, on average. As a caveat, the Google Analytics unique visitor’s numbers are probably an overcount, and the SimilarWeb values may be more in line with reality. As an easier ‘rule of thumb’, we suggest using a 20% adjustment (i.e., increase SimilarWeb numbers) for both metrics based on the analysis findings above. The realization that these services can be complementary can improve decision-making that relies on KPIs and metrics from website analytics data.

Page 28, 594: Estimating Google Analytics Metrics for Multiple Websites: As shown above, the differences between Google Analytics and SimilarWeb metrics for total visits and unique visitors are systematic (i.e., the differences stay relatively constant), notably for visits and unique visitors. This means, if you have Google Analytics values for one site, you can adjust and use a similar difference for the other websites to get reasonable analytics numbers to those from Google Analytics. This technique is valuable in competitive analysis situations where you compare multiple sites against a known website and want the Google Analytics values for all sites. However, SimilarWeb generally provides conservative analytics metrics compared to Google Analytics, meaning that, if solely relying on this single service, analytics measures may be lower, especially for onsite interactions. So, decisions using these analytics metrics need to include this as a factor.

16. Regarding figures 6-8. They need improvement. What these numbers mean both in vertical and horizontal axes? And especially on the horizontal one. Although the comparison through the line is comprehensible, the rest are not. Also, we can minimize the white space (where it is possible) by eliminating the range of the vertical access.

We thank the reviewer for this suggestion which we now address in this version of the manuscript. Given the request above for scatterplots for the correlations, which was the purpose of these graphs, we have replaced the graphs with scatterplots in this version of the manuscript.

See Figure 4, page 18

See Figure 5, page 19

See Figure 6, page 20

See Figure 7, page 21

17. Line 405. We refer “that these ranked lists can be used for research and other purposes”. Ok but for what other purposes? this is a little be general. Better to be more explicit here and point out the other purposes.

We thank the reviewer for this suggestion which we now address in this version of the manuscript. 

Page 23, 451: The website rankings by each service are significantly correlated, so it seems that these ranked lists can be used for research on analytics, competitive analysis, and analytics calculations for a set of websites, with the caveat highlighted in (18,19). These analyses compare the two services’ precision (i.e., how close measured values are to each other). 

18. Line 414. This citation (118) is related with the messy situation in Scientometrics and has nothing to do with the web analytics of websites. Better find something else, or remove it.

We thank the reviewer for this suggestion which we now address in this version of the manuscript. Following the reviewer’s suggestion, we have removed the offending citation. 

Page 23, 461: No measure or measurement tool is perfect, and web data can be particularly messy. 

19. Line 420. there is no “installed correctly or the same on all the websites”. The script is one. If it is installed, then produces numbers. If it is not, then no numbers. Of course, there are incorrections between the connections of GA with Google Ads, Search Console or other platforms and their produced metrics. But in case of the three metrics that have been used here, there are measured properly or indicated zero values if there is a problem in set up. In addition, if we have doubts about the proper installation of GA, why we do not use the Tag Assistant Legacy of Google as browser extension in our data collection section? This tool identifies errors in analytics installation (check here https://chrome.google.com/webstore/detail/tag-assistant-legacy-by-g/kejbdjndbnbjgmefkgdddjlbokphdefk?hl=en)

We thank the reviewer for raising this issue. Perhaps “installed correctly” presented the incorrect impression, so we have modified the sentence.

Page 23, 469: Furthermore, Google Analytics might have different settings in terms of filtering, such as housekeeping visits from organizational employees that would slant the results. 

20. Line 433-438. Again, regarding the Bounce Rate metric. Well, this is a contradictory justification with the aforementioned definition of Bounce Rate as can be seen in table 2. And if we want to consider the Duration as the third central measurement of Web Analytics, why we choose bounce rate which is at least contentious for many cases in the literature review regarding duration validity? And not choosing the visit duration (SW) and the Avg. Visit Duration (GA) to make a comparison among them? This will eliminate all these doubts about bounce rate validity.

We thank the reviewer for raising this issue concerning bounce rate, which we have changed from Duration to Engagement. As for the reviewer’s comment that visit duration or average visit duration totally addresses the duration is not entirely correct, as any measure of duration will suffer from the ‘no exit point’ issue, so this exit point issue does not only affect bounce rate.

Additionally, we now have added an additional section for analysis of average session duration for the duration metric. 

Page 9, 204: H4: SimilarWeb measures of average session durations for websites differ from those reported by Google Analytics.

See section H4: Measurements of average session duration differ, beginning on page 21, 416

21. Line 525. These two citations on this line. The first one 119, refers issues about the setup errors of GA. However, none of these errors of administrators affect the three metrics that we involve here. For example, if we were involving demographics, then ok, we have validity problems. But none of the statements of Alex Ramadan affect the Total visits, Unique Visitors and Bounce Rate. The other link (citation 119) is broken as a 404 page.

We have removed the sentence from the paragraph in this version of the manuscript as upon review, it was not central to the paragraph’s main topic. 

22. Regarding reference list. Citations 28, 32, 33, 54, 55, 84 are broken or are not working properly.

Thanks for pointing out this issue. Impressed that you checked them all! 

We also tested the links. Of the six you mention, four worked fine for us, and two links did not. 

We now provide updated functional links for those two references or removed them. 

We also verify that every link in the reference listing was functional as of the date that we submitted this manuscript. 

Again, thanks! 

End of Comments/Suggestions. Thank you for this opportunity.

Hey, thank you for the GREAT comments and suggestions! Made us work, but we believe the suggestions and effort to address these suggestions really improved the manuscript! 

Also, the tone of the comments and positive support were really motivating for us to do a good job with the revisions! Again, thanks so much! 

-------------

Reviewer #2: This is a very well-written manuscript. Very easy to read. The material is well-organized.

We thank the reviewer for the positive comment about the manuscript, which we also believe is well-written, easy to read, and well-organized material. 

The manuscript deals with an important problem area: the accuracy of popular website analytics and traffic estimation services (e.g., Google Analytics and SimilarWeb). The manuscript identifies and addresses a research gap: a lack of academic research and interest in studying web analytics.

We thank the reviewer for the positive comments about the research, which we also believe is an important problem, and the manuscript, which we also believe identifies and addresses a research gap: a lack of academic research and interest in studying web analytics. 

To improve, can the authors provide more insight on why there is a lack of attention among academics in currently studying this phenomenon? 

We thank the reviewer for this suggestion which we now address in this version of the manuscript.

Page 7, 142: While few academic studies have examined analytics services, fewer have evaluated the actual analytics numbers; instead, they focus on the more easily accessible (and usually free) ranked lists. Studies are even rarer still on the performance of SimilarWeb, despite its standing and reputation as an industry leader. Scheitle and colleagues (19) attribute this absence to SimilarWeb charging for its service, although the researchers do not investigate this conjecture.

Page 8, 181: Although the questions are conceptually straightforward, they are surprisingly difficult to execute in practice. This difficulty, especially in terms of data collection, may be a compounding factor for the dearth of academic research in the area. 

In the abstract, rather than saying the accuracy of metrics provided by Google Analytics and SimilarWeb will be discussed, provide a short sentence or two that speaks to or describes the accuracy of these metrics. 

We thank the reviewer for this suggestion which we now address in this version of the manuscript.

Abstract, 25: The accuracy/inaccuracy of the metrics from both services is discussed from the vantage of the data collection methods employed. In the absence of a gold standard, combining the two services is a reasonable approach, with Google Analytics for onsite and SimilarWeb for network metrics. 

In the paper, provide more insight on what the impact of SimilarWeb providing conservative traffic metrics compared to Google Analytics actually means in terms of practice. Why should we care? Why is it important to know that SimilarWeb and Google Analytics can be used in a complementary fashion when direct website data is not available? How important is this to know? Elaborate more on the implications of this research.

We thank the reviewer for these suggestions, which we now address in this version of the manuscript.

• Page 27, 576: Use of Google Analytics and SimilarWeb: Findings of our research show that, in general, SimilarWeb results for total visits and number of unique visitors will generally be lower than those reported by Google Analytics, and the correlation between the two platforms is high for these two metrics. So, if one is interested in ranking a set of websites for which one does not have the Google Analytics data, the SimilarWeb metrics are a workable proxy. If one is interested in the actual Google Analytics traffic for a set of websites, one can use the SimilarWeb results and increase by about 20% for total visits and about 40% for unique visitors, on average. As a caveat, the Google Analytics unique visitor’s numbers are probably an overcount, and the SimilarWeb values may be more in line with reality. As an easier ‘rule of thumb’, we suggest using a 20% adjustment (i.e., increase SimilarWeb numbers) for both metrics based on the analysis findings above. The realization that these services can be complementary can improve decision-making that relies on KPIs and metrics from website analytics data. 

• Page 28, 591: Estimating Google Analytics Metrics for Multiple Websites: As shown above, the differences between Google Analytics and SimilarWeb metrics for total visits and unique visitors are systematic (i.e., the differences stay relatively constant), notably for visits and unique visitors. This means, if you have Google Analytics values for one site, you can adjust and use a similar difference for the other websites to get reasonable analytics numbers to those from Google Analytics. This technique is valuable in competitive analysis situations where you compare multiple sites against a known website and want the Google Analytics values for all sites. However, SimilarWeb generally provides conservative analytics metrics compared to Google Analytics, meaning that, if solely relying on this single service, analytics measures may be lower, especially for onsite interactions. So, decisions using these analytics metrics need to include this as a factor. 

There are 143 references included in this paper. This is great, but over the top. I think the references can be reduced to a more significant subset. This would reduced the paper’s word count.

We thank the reviewer for this suggestion. Since this is one of the first research papers in this area, we need to ensure that the literature review is comprehensive, requiring a substantial number of references, along with the many technical references required for a thorough explanation of the two platforms and measures employed.

What impact do the study’s findings have on user-centric, site-centric, and network-centric approaches to web analytics data collection identified earlier in the paper?

We thank the reviewer for this suggestion which we now address in this version of the manuscript.

Page 26,533: Triangulation of Data, Methods, and Services: There seems, at present, to be no single data collection approach (user, site, or network-centric) or web analytics service (including Google Analytics or SimilarWeb) that would be effective for all metrics, contexts, or business needs. Therefore, a triangulation of services, depending on the data, method of analysis, or need, seems to be the most appropriate approach. It appears reasonable that user-centric approaches can be leveraged for in-depth investigation of user online behaviors, albeit usually with a sample. Site-centric approaches can be leveraged for the investigation of users’ onsite behaviors. Network-centric approaches can be leveraged for in-depth investigation of user intersite behaviors.

The USA represents half of the 86 websites studied. News and media content represents 42% of the 86 websites. It would be good to further describe how the this skewed sample affects the findings and interpretation of results.

We thank the reviewer for this suggestion which we now address in this version of the manuscript.

Page 28, 610: An additional limitation is that a large percentage of the sites used in this research are content creation sites based in the U.S.A., which might skew user behavior. Other future research involves replication studies with different sets of websites, other website analytics services, other metrics, and analysis of specific website segments based on type, size, industry vertical, or country (i.e., China being a critical region of interest).

-------------

Reviewer #3: The authors conducted a comparison between Google Analytics and SimilarWeb based on analytics metrics data. The results provide both theoretical and practical implications. The paper is clearly organized and well written. With some minor improvements this piece is worth publishing, and I have a few specific suggestions.

We thank the reviewer for both the accurate summary of the research and the positive comments about the research (which we also believe provides both theoretical and practical implications) and the manuscript, which we also believe is clearly organized and well written. We also believe the research manuscript is worth publishing. 

Concerning the few specific suggestions, we believe we have addressed both the spirit and the specifics of the reviewers’ comments in the manuscript’s current version, as outlined below.

First, the authors need to justify their selection of total visits, unique visitors, and bounce rates as the three metrics. Why excluding other common metrics such as time on site/page?

We thank the reviewer for this suggestion which we now address in this version of the manuscript. Additionally, we now include time on-site analysis in this version of the manuscript. 

Page 8, 187: To investigate this research objective, we focus on four core web analytics metrics – total visits, unique visitors, bounce rate, and average session duration – which we define in the methods section. Although there is a lengthy list of possible metrics for investigation, these four metrics are central to addressing online behavioral user measurements, including frequency, reach, engagement, and duration, respectively. We acknowledge that there may be some conceptual overlap among these metrics. For example, bounce rates are sessions with an indeterminate duration, but average session duration also provides insights into user engagement. Nevertheless, these four metrics are central to the web analytics analysis of nearly any single website or set of websites; therefore, they are worthy of investigation. In the interest of space and impact of findings, we focus on these four metrics, leaving other metrics for future research.

Second, all three hypotheses are supported, but how does this help evaluate the accuracy of the two analytics services? I think it impossible to indicate which one is more accurate given the significant differences between them in terms of the three metrics.

We thank the reviewer for this suggestion which we now address in this version of the manuscript.

Abstract, 25: The accuracy/inaccuracy of the metrics from both services is discussed from the vantage of the data collection methods employed. In the absence of a gold standard, combining the two services is a reasonable approach, with Google Analytics for onsite and SimilarWeb for network metrics.

Page 9, 196: Given that Google Analytics uses site-centric website data and SimilarWeb employs a triangulation of datasets and techniques, we would reasonably expect values would differ between the two. However, is it currently unknown how much they differ, which is most accurate, or if the results are correlated. Therefore, because Google Analytics is the de facto industry standard for websites, we use Google Analytics measures as the baseline for this research.

Page 23,460: Although one might lean toward considering metrics reported by Google Analytics as the ‘gold standard’ for website analytics (and justifiably so in many cases), it is also known within the industry that Google Analytics has tracking issues in some cases. Also, a reportedly high percentage of Google Analytics accounts are incorrectly set up (118–121), skewing the measuring in some cases. There are also contexts where other analytics methods might be more appropriate. Google Analytics relies on one data collection approach: basically, a cookie and tagging technique. There are certainly cases (e.g., cleared cookies, incognito browsing) when this method is inaccurate (e.g., unique visitors). Furthermore, Google Analytics might have different settings in terms of filtering, such as housekeeping visits from organizational employees that would slant the results. Therefore, these issues result in issues with Google Analytics being seen as the ‘gold standard.’ 

See Practical Implications section, pages 27-28

Finally, I suggest that the authors improve their discussion section by providing more insights into the causes for the differences between Google Analytics and SimilarWeb.

We thank the reviewer for this suggestion which we now address in this version of the manuscript.

See Discussion section, pages 22-26

A minor problem - I’m confused by the statement “The techniques used by SimilarWeb are similar to the techniques of other traffic services, such as Alexa, comScore, SEMRush, Ahrefs, and Hitwise.” (Page 5, Line 100). While Alexa and comScore user-centric, SEMRush, Ahrefs, and Hitwise are network-centric. Why “similar”? What are the “techniques”?

Sorry for this confusion. We have now expanded the sentence with a clarifying phrase in this version of the manuscript.

Page 6, 198: The techniques used by SimilarWeb are similar to the techniques of other website analytics services, such as Alexa, comScore, SEMRush, Ahrefs, and Hitwise, in the employment of user, site, and/or network data collection.

---

## [Decision Letter · Decision Letter 1]

26 Apr 2022

Measuring user interactions with websites: A comparison of two industry standard analytics approaches using data of 86 websites

PONE-D-21-03616R1

Dear Dr. Jansen,

We’re pleased to inform you that your manuscript has been judged scientifically suitable for publication and will be formally accepted for publication once it meets all outstanding technical requirements.

Kind regards,

Hussein Suleman, PhD

Academic Editor

PLOS ONE

Additional Editor Comments (optional):

Reviewers' comments:

Reviewer's Responses to Questions

**Comments to the Author**

1. If the authors have adequately addressed your comments raised in a previous round of review and you feel that this manuscript is now acceptable for publication, you may indicate that here to bypass the “Comments to the Author” section, enter your conflict of interest statement in the “Confidential to Editor” section, and submit your "Accept" recommendation.

Reviewer #1: All comments have been addressed

Reviewer #2: All comments have been addressed

2. Is the manuscript technically sound, and do the data support the conclusions?

Reviewer #1: Yes

Reviewer #2: Yes

3. Has the statistical analysis been performed appropriately and rigorously? 

Reviewer #1: Yes

Reviewer #2: Yes

4. Have the authors made all data underlying the findings in their manuscript fully available?

Reviewer #1: Yes

Reviewer #2: Yes

5. Is the manuscript presented in an intelligible fashion and written in standard English?

Reviewer #1: Yes

Reviewer #2: Yes

6. Review Comments to the Author

Reviewer #1: A big "Well Done" to the authors.

The have addressed all the suggestions and comments to improve the quality of the paper.

Each kind of suggestion has been taken into consideration, while correcting all the things that probably confuse the forthcoming readers.

One of the most crucial aspect (the bounce rate involvement) has also been overhauled with clarity and in a well-organized way. This is a tremendous effort of yours. One step further, you kept the bounce rate and add one more metric.

I think that the paper now stands sufficiently and it constitutes a scientific work that holistically improves the Web Analytics research topic.

Reviewer #2: The authors have adequately address all prior concerns that I (Reviewer #2) previously raised. They have also enriched the quality of the manuscript by adequately addressing all the detailed concerns outlined previously by Reviewer #1. The paper is ready for publication.

7. PLOS authors have the option to publish the peer review history of their article (what does this mean?). If published, this will include your full peer review and any attached files.

Reviewer #1: **Yes: **Prof. Georgios A. Giannakopoulos

Reviewer #2: No

---

## [Editor Report · Acceptance letter]

13 May 2022

PONE-D-21-03616R1 

Measuring user interactions with websites: A comparison of two industry standard analytics approaches using data of 86 websites 

Dear Dr. Jansen:

I'm pleased to inform you that your manuscript has been deemed suitable for publication in PLOS ONE. Congratulations! Your manuscript is now with our production department. 

Kind regards, 

on behalf of

Dr. Hussein Suleman 

Academic Editor

PLOS ONE